# New science of climate change impacts on agriculture implies higher social cost of carbon

Frances C. Moore[1], Uris Baldos[2,3], Thomas Hertel[2,3,4] & Delavane Diaz[5]

Despite substantial advances in climate change impact research in recent years, the scientific basis for damage functions in economic models used to calculate the social cost of carbon (SCC) is either undocumented, difficult to trace, or based on a small number of dated studies. Here we present new damage functions based on the current scientific literature and introduce these into an integrated assessment model (IAM) in order to estimate a new SCC. We focus on the agricultural sector, use two methods for determining the yield impacts of warming, and the GTAP CGE model to calculate the economic consequences of yield shocks. These new damage functions reveal far more adverse agricultural impacts than currently represented in IAMs. Impacts in the agriculture increase from net benefits of \$2.7 ton$^{-1}$ $CO_2$ to net costs of \$8.5 ton$^{-1}$, leading the total SCC to more than double.

[1] Department of Environmental Science and Policy, University of California Davis, 2140 Wickson Hall, One Shields Avenue, Davis, CA 95616, USA. [2] Global Trade Analysis Project Purdue University, 403 West State Street, West Lafayette, IN 47907, USA. [3] Department of Agricultural Economics, Purdue University, 403 West State Street, West Lafayette, IN 47907, USA. [4] Purdue Climate Change Research Center, 203 S. Martin Jischke Drive Gerald D. and Edna E. Mann Hall, Suite 105, West Lafayette, IN 47907, USA. [5] Department of Management Science and Engineering, Stanford University, Stanford, CA 94305, USA. Correspondence and requests for materials should be addressed to F.C.M. (email: fmoore@ucdavis.edu)

Climate science has advanced significantly in the past 20 years so that our understanding of the physical consequences of greenhouse gas emissions is now well established[1]. The biophysical effects of changes in temperature and rainfall on, for example, ecosystems, agricultural yields, and sea-level rise are also increasingly well understood. However, this new science is not reflected in some of the highly influential economic models currently used to determine the social cost of carbon (SCC)—a measure of the total damages from an additional ton of $CO_2$ emissions used to quantify the benefits of emissions reduction. In most cases, the scientific basis for damage functions (reduced-form expressions for how climate change affects economic welfare) in these models is undocumented, tautological (based on damages from previous versions of the models), or dates from between 10 and 20 years ago and therefore may have been superseded by more recent results[2].

The lack of a current empirical basis for integrated assessment model (IAM) damage functions is not just an academic question because the SCC has been formally adopted by the U.S. government to quantify the benefits of $CO_2$ mitigation in cost–benefit analysis[3]. Regulations with benefits totaling over $1 trillion have used the SCC in cost–benefit analysis[4]. Increasingly, it is also being used at the state level: recent rulings in California, New York, and Minnesota all require use of the SCC in analysis of climate and energy regulations[5–7]. Therefore, the value of the SCC is a relevant consideration for long-term planning in industry and government, and yet the extent to which it would change if more recent scientific knowledge were incorporated is largely unknown. Many recent commentaries on the state of climate change economics have identified improving the empirical basis of IAM damage functions as high-priority area for future work[2, 8–12].

A large and growing body of science has dramatically improved our knowledge of the social and economic risks posed by climate change and therefore provides an opportunity to substantially improve the empirical basis of the damages underlying the SCC[12–15]. However, translating the literature on biophysical climate change impacts into damage functions is not straightforward. For each sector, individual scientific studies must be aggregated and translated into a consistent set of global impacts. Then, because damage functions parameterize how economic welfare changes with temperature, the economic value of these biophysical impacts must be assessed, which might involve substantial economic modeling. Finally, new damage functions must be introduced into an IAM in order to examine the effect on the SCC. The complexity and interdisciplinary nature of this process may be part of the reason damage functions in IAMs have lagged the science of climate change impacts.

This paper focuses on the agricultural sector, connecting a current and comprehensive review of the biophysical science of the impacts of climate change on yields to the SCC. Agriculture is an important sector for climate change damages because it is both

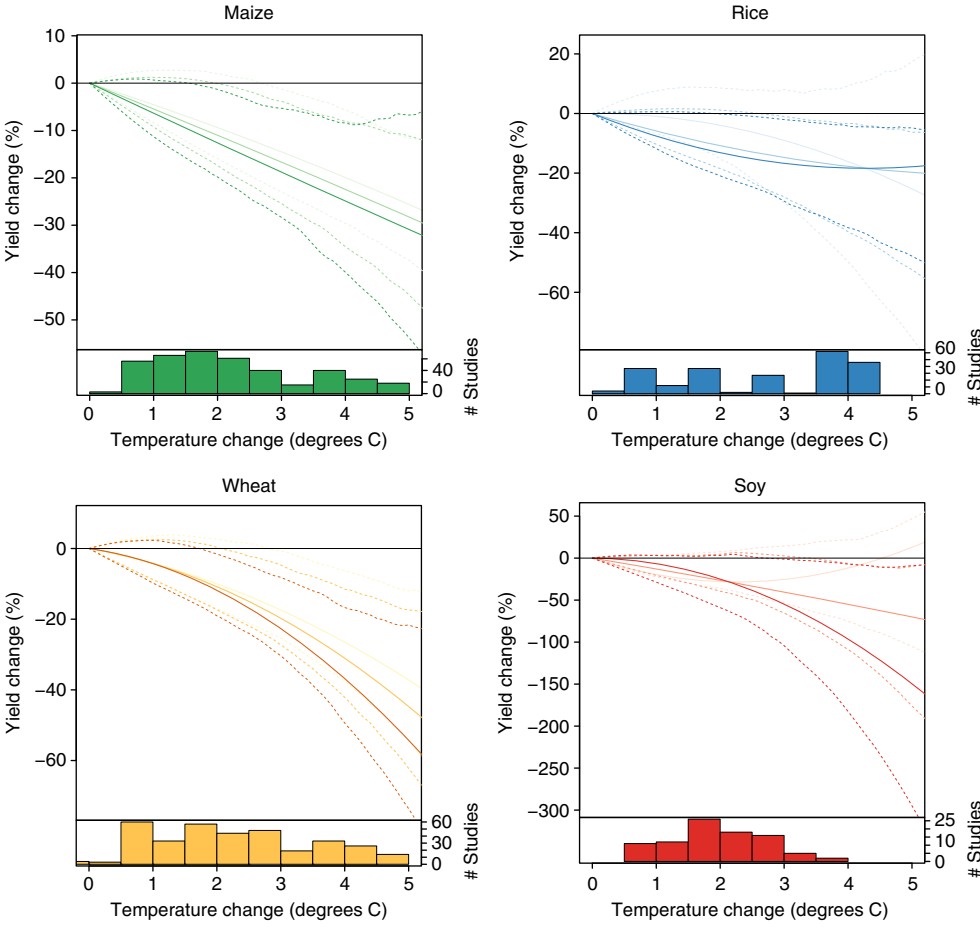

**Fig. 1** Impacts of temperature change on yields of four major crops. Based on a meta-analysis of 1010 point-estimates from 56 studies (see Methods section). Darkest, middle, and lightest lines show responses at the 75th, 50th, and 25th quantiles of baseline growing-season temperature, respectively. Dashed lines show the 95% confidence interval based on 750 block bootstraps, blocking at the study level. Plotted response curves are for temperature only and do not include $CO_2$ fertilization or adaptation. Temperature changes are relative to a local 1995–2005 baseline. The histograms show the number of observations by crop and level of warming used to estimate the response functions. In subsequent analyses, yield losses >100% are set to losses of 99%

directly affected by climate change and has critical implications for future food security and social welfare. We start with the large agronomic literature on how climate change affects crop yields based on a meta-analysis published by Chalinor et al[16]. and used to support conclusions in the IPCC 5th Assessment Report[17]. We present a new analysis of this database that we use to aggregate these results to the global scale. We then compare our results with those of the Agricultural Model Intercomparison and Improvement Project (AgMIP) published by Rosenzweig et al[18]. Using the predicted yield under climate change, derived both from the meta-analysis and from AgMIP as inputs to the GTAP computable general equilibrium (CGE) model, we estimate the economic consequences of these changes. Finally, we parameterize two new damage functions based on the CGE results and incorporate them into one of the most widely used IAMs in order to examine implications for the SCC.

This approach exemplifies several principles that we believe can provide important guidance in updating damage functions in other sectors in the future. Our meta-analysis is related to the findings of the food security chapter of the IPCC 5th Assessment Report. Where possible, tying damage functions to the IPCC has several benefits: findings are updated on a moderately regular basis (every 7 years), are assembled and reviewed by impact experts within each field, and are formally accepted as fact by governments involved in the process. Both the meta-analysis and the AgMIP damage functions also use an ensemble of models. Findings in climate and, increasingly, agricultural modeling have shown that use of multi-model ensembles tend to outperform any individual model[1, 19]. Therefore, averaging multiple model outputs should lead to more reliable damage functions and better uncertainty quantification than trying to pick preferred models. Finally, this is an end-to-end analysis in which analysis of the underlying biophysical impacts literature to produce global, sector-wide response functions, modeling of the economic responses to those biophysical changes using a state of the art, open-source CGE model to produce economic damages, and introduction of these damages into an IAM and the resulting effect on the SCC are all documented within a single study. This means our damage functions and the changes we identify in the SCC have a clear and traceable connection to the underlying science that is both comprehensive and up-to-date, in contrast to most current IAM damage functions[2]. This approach is also consistent with the National Academy of Sciences report on calculating the SCC, which recommended that damage functions should be based on the current, peer-reviewed literature on climate impacts, have uncertainties that are characterized and quantified where possible, and be transparent, well-documented and reproducible[2].

## Results

**Estimating the global yield response to climate change.** The first step in our analysis involves aggregating the large volume of research on how climate change affects crop yields. We do this through a meta-analysis of 1010 published estimates of yield response to changing climate conditions, including both statistical and process-based studies, using a database complied for the IPCC 5th Assessment Report[16, 17]. Figure 1 shows the temperature response functions we derive for the four most important food crops (see Methods section). The effect of higher temperatures on yields is negative for all crops in almost all locations. The interaction between the effects of warming and current growing-season temperature is in the expected direction, with warming consistently more damaging in places that are already hot. However, for wheat and maize (and soybeans at low levels of warming) this effect is not particularly large.

The effects of other variables are shown in Supplementary Information (SI), Supplementary Table 1 and Supplementary Fig. 1. $CO_2$ has a positive effect on crop yields, with an estimated increase of 11.5% ($C_3$) and 8.7% ($C_4$) for a doubling of $CO_2$ from preindustrial levels. This is very close to estimates from experimental field studies for $C_3$ crops but is somewhat high for $C_4$ crops[20, 21]. The effect of agronomic, on-farm, within-crop adaptations (principally changes in crop variety and planting date (see Methods section)) is small and statistically insignificant. Studies that include agronomic adaptation do, on average, report higher yields than those that do not, but this is almost entirely captured by an adaptation intercept term rather than the interaction with change in temperature. This suggests that changes described as adaptations to climate change in the studies underpinning our meta-analysis would provide similar yield benefits with and without climate change and are therefore not true climate adaptations as conventionally defined[22]. In results that follow we include only the true climate adaptation effect for all crops, but we find this to be small (Supplementary Table 1). Note that this statement refers only to the on-farm, within-crop agronomic adaptations captured by the studies that support the meta-analysis. Additional economic adaptations such as crop switching, increasing production intensity, substituting consumption, or adjusting trade relationships are captured in the GTAP model (Supplementary Table 2).

Our continuous response functions are extrapolated globally using maps of baseline temperature and the spatial pattern of global temperature change in order to estimate yield changes for different levels of global warming (Methods, Supplementary Figs 2–5). We compare these results to those from the AgMIP Global Gridded Crop Model Intercomparison (GGCMI), the one other source of global, multi-crop, multi-model yield changes (Supplementary Fig. 6)[18]. Our preferred results focus on the set of models within the AgMIP ensemble that explicitly represent nitrogen stress (see Methods section) though in the SI we also present results using the full ensemble. The two approaches for estimating yield impacts differ substantially in the kinds of spatial heterogeneity in the yield response to warming and $CO_2$ that are captured. The GGCMI results explicitly account for spatial variation resulting from soil type, irrigation, baseline temperature, and (in models representing nitrogen stress) nutrient limitations. The meta-analysis deliberately smooths out most of this heterogeneity in order to more precisely estimate a common response function, preserving only the heterogeneity resulting from different baseline temperatures. (See also Supplementary Table 3 for additional discussion on sources of spatial heterogeneity).

With the exceptions of soybeans, our point-estimates show substantial areas of agreement (within 10 percentage points of the GGCMI). A major difference is that our latitudinal variation in yield impacts tends to be more modest than in the GGCMI ensemble, leading our meta-analysis results to be more optimistic in tropical areas and more pessimistic at higher latitudes. At the global scale, these cancel to some degree so that differences in global, production-weighted yield changes between the two methods are smaller and, due to large uncertainties involved, statistically indistinguishable from each other (Supplementary Fig. 7). The largest area of disagreement is for wheat yields—with the AgMIP ensemble showing global productivity gains for 2 and 3 degrees of warming and the meta-analysis showing substantial losses.

**Economic consequences of yield impacts.** We use gridded yield changes for the four major crops for 1–3 °C based on both our meta-analysis and the GGCMI ensemble average as input to the

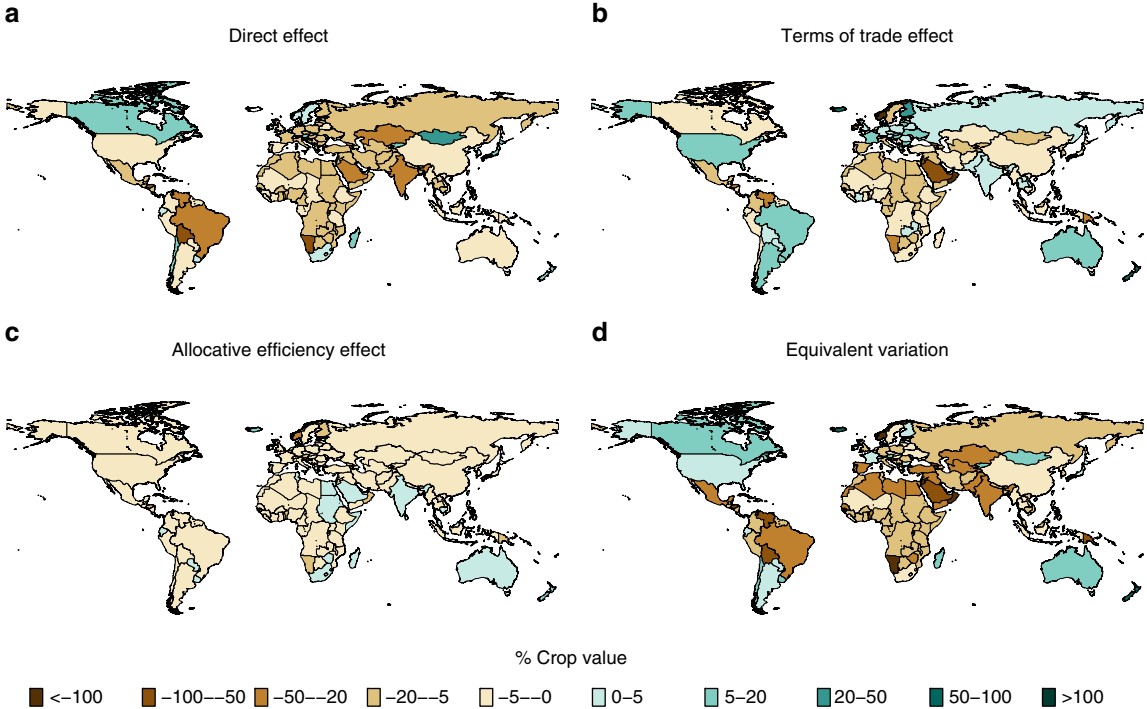

**Fig. 2** Welfare changes from 3 °C of global average warming. Changes are relative to a 1995–2005 global average baseline and use yield changes based on the meta-analysis results shown in Fig. 1: **a** the direct technical effect of climate change on agricultural productivity; **b** terms of trade effects; **c** the allocative efficiency effect; and **d** total welfare change reported as equivalent variation. Results are based on yield changes that include adaptation and the $CO_2$ fertilization effect for $C_3$ crops but not for maize. Welfare changes are normalized by the value of production of the affected crops (maize, rice, wheat, and soybeans)

GTAP CGE model (see Methods section). Our aggregation of the version 9 GTAP data base results in a model that solves for equilibrium prices, consumption, production, and bilateral trade flows of 14 commodities (of which 9 are in the agricultural sector) across 140 regions (see Methods section)[23, 24]. This step is necessary because IAM damage functions parameterize how economic welfare changes with global temperature and, beyond the direct productivity effects, the relationship between yield and welfare changes is not straightforward. This complexity derives from the presence of additional impacts on a nation's terms of trade (changes in the relative prices of a region's exports and imports) and an allocative efficiency effect (interactions between changing production, consumption, and trade patterns and existing market distortions)[25]. The sum of these three is the total regional welfare change, reported here as real income.

Figure 2 shows regional welfare changes associated with 3 °C of warming based on the meta-analysis results, normalized by the current value of the four crops being modeled. At this level of warming, total effects on welfare (Fig. 2d) are negative in most areas, primarily driven by the direct productivity effect of climate change on agriculture (Fig. 2a). However, terms-of-trade effects (Fig. 2b) are important in determining the distribution of global welfare gains across individual regions, comprising >50% of total welfare change in some regions (Supplementary Fig. 8). Because world crop prices are increasing in this scenario (Supplementary Fig. 9), it is the net agricultural exporters that tend to gain from changing terms of trade (e.g., Australia, Argentina, USA), while net importers lose (e.g., Mexico, the Middle East, North Africa). Supplementary Fig. 10 shows the same welfare decomposition but based on yield changes from the GGCMI ensemble (Supplementary Fig. 11 shows results for the full ensemble, including models that do not represent nitrogen stress). Both sets of results show welfare declines in south Asia, sub-Saharan Africa, Brazil,

Mexico, and China. However, the GGCMI results show much wider welfare gains from productivity improvement in higher latitudes. In addition, price changes are more variable (with an increase in maize and soy prices but decreases in wheat and rice prices), so that the terms-of-trade effects are smaller and are distributed differently between importers and exporters of different crops. Note that our results differ from Nelson et al[26]., who also used the GGCMI ensemble as input to a range of general- and partial-equilibrium models. A major reason why our results differ from theirs is that they did not include $CO_2$ fertilization, which is included in our yield shocks.

**Implications for the social cost of carbon.** To create new damage functions, we take welfare changes for 1–3 °C of average global warming and aggregate up to the 16 geographic regions used in the FUND model[27]. Figure 3 shows our damage functions estimated from both the meta-analysis and GGCMI yield responses, compared to the existing agricultural sector damage functions in FUND (see also Supplementary Fig. 12 and Supplementary Table 4). Agriculture in FUND shows benefits in all regions for warming < 3 °C. This is a result of both a direct positive effect of moderate amounts of warming on yields for all regions and the $CO_2$ fertilization effect. In contrast, our meta-analysis results show almost universal negative welfare changes for warming beyond 2 °C that in many cases are very large. Current FUND welfare impacts are almost entirely at or beyond the upper bound of our 95% confidence intervals (Fig. 3). Results for the preferred AgMIP GGCMI ensemble fall in between these two cases. In some regions (Japan+Korea and the Middle East), they closely track existing FUND damages. In other regions (Central America, USA, South Asia), they show welfare declines at 2 and 3° of warming that are similar to our meta-analysis results.

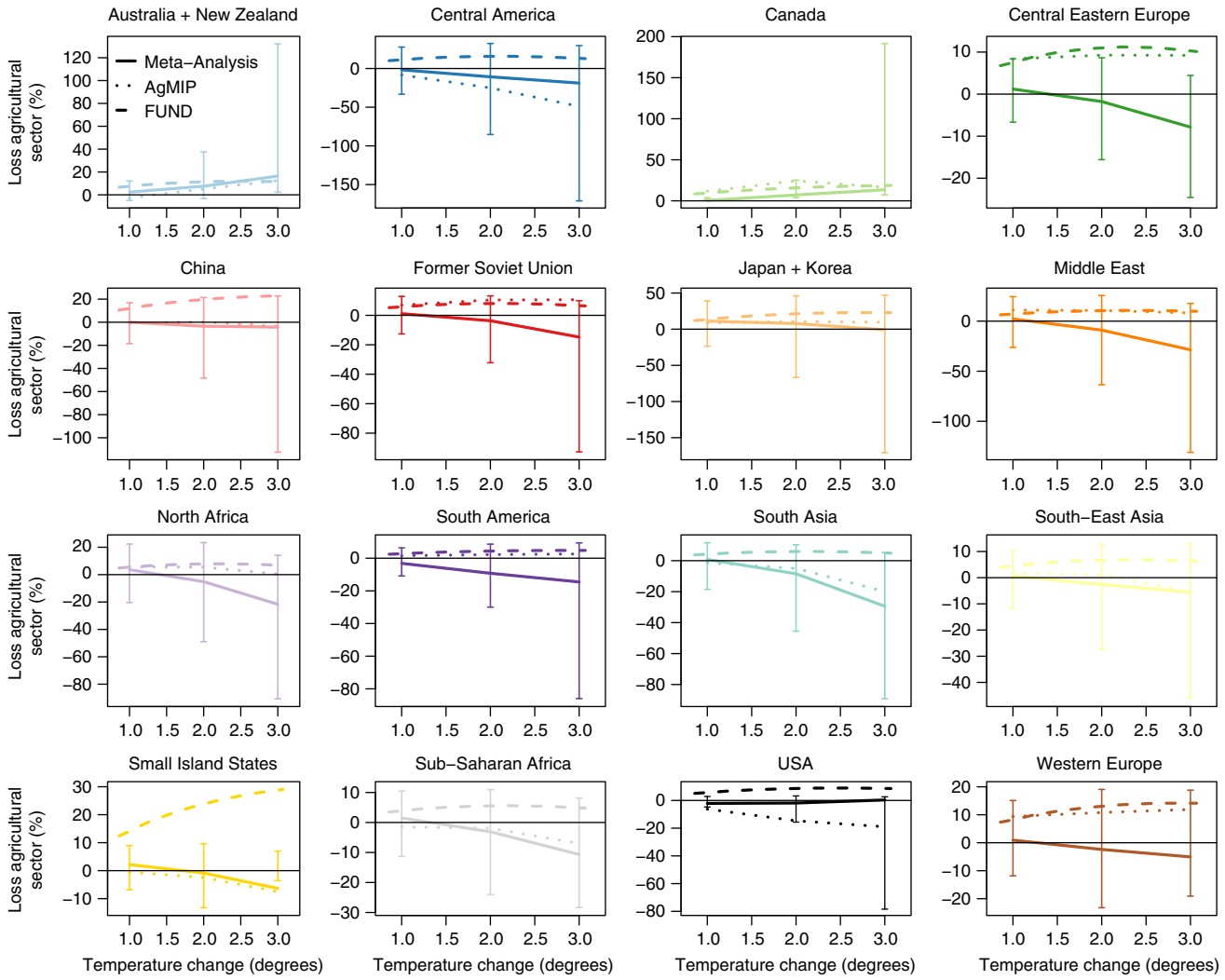

**Fig. 3** Three agriculture-sector damage functions for each of the 16 FUND regions. Solid lines are from meta-analysis results, dotted lines are AgMIP results, and dashed lines are the existing FUND damage functions. Error bars show the damage functions based on the 2.5th and 97.5th quantiles of the meta-analysis results. Temperature changes are global averages and are relative to a global average 1995–2005 baseline. A version of the figure excluding error bars, which allows differences in the point-estimates to be more easily distinguished, is given in Supplementary Fig. 12

In order to investigate the importance of parameterization of the economic model used to calculate damages in driving the regional damage functions, we perform a systematic sensitivity analysis of key parameters within GTAP (see Methods section). Variation in economic welfare changes associated with these changes is shown in Supplementary Fig. 13 and is very small and is dwarfed by the uncertainty in biophysical productivity shocks shown in the error bars in Fig. 3.

We use an IAM damage module based on the FUND model and substitute our new agricultural damage functions into the agricultural sector to calculate how the SCC changes as a result of this more up-to-date science and new economic modeling (see Methods section). Because total damages in FUND are an additive sum over regions and sectors, the SCC can be decomposed into its constituent parts. Figure 4 summarizes the results of this analysis, using a 3% discount rate. Additional results based on a 2.5 and 5% discount rate are given in Supplementary Table 5. Currently, agriculture in FUND contributes a benefit of $2.7 ton$^{-1}$ $CO_2$ toward the SCC. In contrast, damage functions based on both the preferred AgMIP GGCMI ensemble and the meta-analysis show net costs of $3.5 and $8.5 ton$^{-1}$, respectively (95% confidence interval based on the spread

in yield impacts for the meta-analysis is −$0.6 to $33.3 ton$^{-1}$). This difference has a substantial effect on the SCC: although the FUND model represents damages from 14 impact sectors, only a few key sectors—agriculture, cooling, and heating—contribute substantively to the SCC[28]. Updating the damage functions in the agriculture sector alone increases the SCC from $8.6 to $14.8 (AgMIP) and $19.7 (meta-analysis) ton$^{-1}$ $CO_2$, increases of 72 and 129%, respectively.

Sensitivity of results to alternative methodological assumptions are shown in Supplementary Table 5, which gives the SCC under alternative discount rates, an alternative method of extrapolating beyond 3 °C, and using the full AgMIP ensemble instead of the preferred set of models that explicitly represent nitrogen stress. Although discount rates have a large effect on the SCC, they scale all results and so do not alter the finding that updated damage functions imply a higher SCC. The largest sensitivity of results is around including the full ensemble of process-based crop models in the AgMIP GGCMI. Adding models that do not represent nitrogen stress substantially reduces the estimated impact of climate change on agriculture and leads to an SCC ($9.6 ton$^{-1}$ with a 3% discount rate) only slightly greater than the current FUND estimate. This difference is due to the presence of large

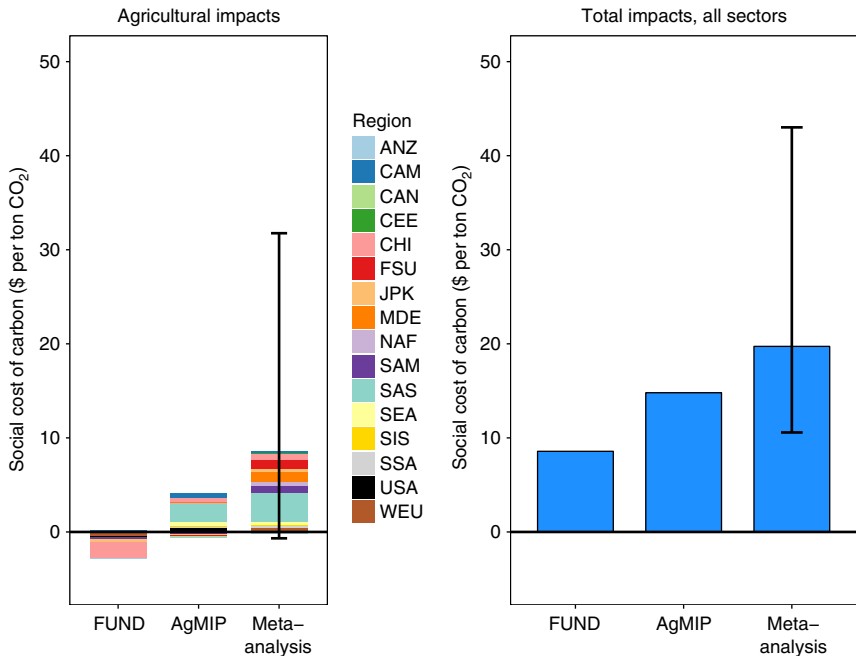

**Fig. 4** Changes in the social cost of carbon resulting from new damage functions in the agricultural sector. **a** Agricultural impacts only, decomposed by geographic region (see Supplementary Table 4 for region definitions). **b** The total social cost of carbon (SCC), keeping damages in all non-agricultural sectors fixed. Results are based on a business-as-usual emissions scenario and a 3% discount rate. Uncertainty bars give the SCC resulting from the 95% confidence interval of yield response parameter estimates from the meta-analysis

productivity gains in many parts of the world (Supplementary Fig. 11a), a product particularly of the LPJ-GUESS and GAEZ-IMAGE models[18].

## Discussion
Here we show that the current science of climate change impacts on agriculture, combined with up-to-date economic modeling, implies larger damages to the sector than currently represented in models used to calculate the SCC. In contrast to existing regional damage functions, which show benefits in every region up to at least 3 °C of warming, we find potential for welfare declines even at much lower levels of warming. Though the range of possible effects of climate change on yields is substantial, our finding that the SCC should be increased is robust to this variation, as well as to uncertainties relating to the discount rate, economic modeling, and extrapolation of the damage function.

Of the three IAMs used to estimate the SCC, only FUND explicitly represents the agricultural sector, so that has been the focus of comparison in this paper. However, agricultural impacts are a part of damages in the two other models. In PAGE (2009 version) agricultural impacts are represented within the market impacts damage function and in DICE (2013 version) they are in the non-sea level rise damage function[29, 30]. Given the same socio-economic assumptions and climate model used to calculate SCC values reported in the previous section, these damage functions give SCC values of \$6.6 ton$^{-1}$ and \$18.9 ton$^{-1}$, respectively (3% discount rate)[28]. In the case of PAGE, market impacts are substantially smaller than agricultural damages estimated using the meta-analysis (\$8.5 ton$^{-1}$), implying either that there are large off-setting benefits in other market sectors or that the empirical basis for the market impacts damage function may need to be reviewed. In the case of DICE, our results imply that agricultural impacts make up between 19% (AgMIP) and 45% (meta-analysis) of non-sea level rise damages.

Governments are currently relying on IAMs to evaluate climate and energy policy and these models have already come under

legal scrutiny as a result[31, 32, 5]. It is therefore important, from both a regulatory and an academic perspective, that the representation of damages reflect current scientific consensus on impacts in a timely and transparent manner. Here we have shown that improving the empirical basis of just one sector, agriculture, results in a large increase in the SCC. In addition, we have demonstrated the potential of an end-to-end analysis directly linking the biophysical impacts of climate change to economic welfare and ultimately the SCC. Damage functions resulting from this approach are more clearly tied to underlying science and can be easily updated in light of future findings. This approach, which can also be extended to other sectors, therefore represents an essential step in maintaining and improving the integrity of IAM results going forward.

## Methods
**Meta-analysis of yield response to climate change**. The yield–temperature response functions used in this paper are derived from a database of studies estimating the climate change impact on yield compiled for the IPCC 5th Assessment Report[17], also described in a meta-analysis by Challinor et al.[16]. Methods for sampling the literature and criteria for inclusion are described in Challinor et al.[16]. as a "broad and inclusive literature search" combined with quality-control procedures documented in the Supplemental Information of that paper. In this study, we focus on four major crops—maize, rice, wheat, and soybeans. The bulk of the scientific literature on yield response to temperature relates to these crops, which collectively account for about 20% of the value of global agricultural production, 65% of harvested crop area, and nearly 50% of calories directly consumed[33]. For the four crops, the database contains 1010 observations (344, 238, 336, and 92 for maize, rice, wheat, and soybeans, respectively) from 56 different studies (many studies report multiple yield changes for different crops, different locations, different levels of temperature change, or different assumptions about adaptation). The studies include 8 empirical studies and 48 process-based studies, published between 1997 and 2012. Supplementary Figs 14 and 15 show the geographic coverage of production areas within the database and the distribution of publication dates. Of the 1010 data points, 451 are reported as including some form of on-farm, within-crop, agronomic adaptation. The vast majority of these adaptations involve adjusting either planting date (10%) or cultivar (12%) or both (44%). Recognizing the existence of a more recent and possibly more systematic literature review for wheat yields, we perform a robustness check where we incorporate additional results identified in Wilcox and Makowski[34]. This substantially increases

the number of observations for wheat but does not affect our estimated response curve (Supplementary Fig. 16).

We merge this database with information on baseline growing-season temperature for each data point. To do this, each data point was assigned to a country. For the 14% of studies looking at more than one country, the country assigned was the one with the highest production of the relevant crop. Average baseline growing-season temperatures were calculated using planting and harvest dates from Sacks et al[35]. and gridded monthly temperatures for 1979–2013 from the Climate Research Unit[36]. These were averaged to the country level using year 2000 crop production weights from Monfreda et al[37].

The response functions are jointly estimated from the point-estimates in the database using a multi-variate:

$$
\begin{aligned}
\Delta Y_{ijk} = & \beta_{1j}\Delta T_{ijk}*Crop_j + \beta_{2j}\Delta T_{ijk}^2*Crop_j + \beta_{3j}\Delta T_{ijk}*Crop_j*\overline{T}_{jk} \\
& + \beta_{4j}\Delta T_{ijk}^2*Crop_j*\overline{T}_{jk} + \beta_5 f_1\big(\Delta CO_{2ijk}\big)*C_{3j} \\
& + \beta_6 f_2\big(\Delta CO_{2ijk}\big)*C_{4j} + \beta_7\Delta P_{ijk} + \beta_8\Delta T_{ijk}*Adapt_{ijk} + \beta_9 Adapt_{ijk} + \varepsilon_{ijk}
\end{aligned}
\tag{1}
$$

where $\Delta Y_{ijk}$ is the change in yield from point-estimate $i$ for crop $j$ in country $k$ (in %). $\Delta T_{ijk}$, $\Delta CO_{2ijk}$ and $\Delta P_{ijk}$ are the changes in temperature (in degree C), $CO_2$ concentration (in parts per million (ppm)), and rainfall (in percent) for point-estimate $ijk$, $\overline{T}_{jk}$ is the baseline growing-season temperature for crop $j$ in country $k$, $C_{3j}$ and $C_{4j}$ are dummy variables indicating whether crop $j$ is $C_3$ or $C_4$, and $Adapt_{ijk}$ is a dummy variable indicating whether the point-estimate includes any on-farm adaptation. Eq. 1 is estimated using an ordinary least squares regression.

Uncertainty in the parameters is estimated through 1500 block bootstraps, with blocks defined at the study level, allowing for possible correlation between point-estimates from the same study. Error bars reported throughout the paper are based on the 2.5th and 97.5th quantiles of the bootstrapped distribution. This treatment of the errors does assume independence between studies, which may be questionable if the same model is used in multiple studies. In total, 28 models, made up of 17 process-based model families (i.e., treating CERES-maize, CERES-rice, and CERES-wheat as a single model) and 11 statistical models, are used in the 56 studies. Supplementary Fig. 17 shows response curves with standard errors based on a model block bootstrap as a robustness check. These are qualitatively similar to the error bars shows in Fig. 1, particularly for warming <3 °C that is the focus of the economic analysis, suggesting the study block bootstrap is capturing the bulk of residual covariance. All error bars reported in the paper show confidence intervals rather than prediction intervals. This is appropriate since the relevant uncertainty is in the expected response of yield to temperature change, which is given by confidence intervals.

There are a number of important things to note about this specification shown in Eq. 1. First, the impacts of temperature are modeled as crop-specific quadratics ($\beta_{1j}$ and $\beta_{2j}$ terms), allowing the effects of warming to vary by crop. In addition, the effects of warming are allowed to vary with baseline growing-season temperature ($\beta_{3j}$ and $\beta_{4j}$ terms), capturing the intuition that the impacts of a 1 °C warming should be different in a cold location than in a hot location.

Second, there is no intercept term, thereby forcing response functions without adaptation through the origin. This is consistent with the expected functional form of a climate damage function, which should have no impacts if there are no changes in climate variables. However, we include an intercept for studies that do include adaptation ($\beta_9$). This is prompted by the observation that, in many studies, 'adaptation' is represented by changing management practices that would improve yields even in the current climate, such as adoption of improved varieties or increasing fertilizer or irrigation inputs[22]. Failing to include an adaptation intercept in this context will lead to an overestimation of the potential of these kinds of changes to reduce the negative impacts of a warming climate. This adaptation intercept is subtracted in our estimates of the effect of climate on yield to produce an adjusted damage function that goes through the origin. (In other words, we calculate the effect of a change in temperature of $X$ on yields to be the yield change predicted from Eq. 1 for a temperature change of $X$ minus the yield change predicted for a temperature change of zero (i.e., $\beta_6 \, Adapt_{ij}$)). The true effect of adaptation is the interaction with temperature change, given by the $\beta_8$ term in Eq. 1, which is included in all subsequent analyses. This term captures the effect of management changes that are not beneficial today but that will become beneficial under a changed climate, the standard definition of adaptation.

Finally, the functions $f_1()$ and $f_2()$ are concave, allowing for a declining marginal effect of $CO_2$, consistent with a number of field studies[20,38]. Specifically, the function takes the form $f\big(\Delta CO_{2ij}\big) = \frac{\Delta CO_{2ij}}{\Delta CO_{2ij}+A}$ where $A$ is a free parameter set at 100 ppm for $C_3$ crops and at 50 ppm for $C_4$ crops based on a comparison of the $R^2$ across models using multiple possible values. The changes in $CO_2$ are adjusted so that all are relative to a modern baseline of 360 ppm (the most common baseline value for studies included in the analysis).

In addition to Eq. 1, our preferred specification, we investigate the effects of several alternate specifications. Specifically we first investigate whether newer studies (publication date of 2005 or later) give a different temperature response compared to the full sample; second investigate the effect of individual agronomic adaptations, specifically changing cultivar and planting date; third allow the effect of temperature to differ depending on whether the study was a process-based or empirical study; fourth add a $\Delta T_{ijk}^3$ term in the specification; and finally perform

$F$-tests on individual terms within the model. These findings are documented in the SI (Supplementary Figs 18–20, Supplementary Tables 6 and 7). They do not substantially alter our estimates of the yield response to climate change.

**Gridded yield changes**. After estimating Eq. 1, we developed global gridded yield change scenarios for the four major crops (maize, wheat, rice, and soybeans). Although IAM damage functions are typically based on global temperature changes, it is important to account for the fact that local warming may differ significantly from global warming in estimating impacts. Local yield impacts will depend on local temperature changes, which scale in a predictable way with global temperature change. We estimate this scaling using the CMIP5 multi-model ensemble mean for the high emissions scenario RCP 8.5[39]. For each grid cell, we take the change in temperature between a future (2035–2065) and baseline (1861–1900) period and divide by the mean global warming over this time period, giving the pattern scaling relationship between global and local temperature change for each grid cell (Supplementary Fig. 21). For a given increase in global mean temperature, warming is larger over land than over the ocean and at high latitudes compared to the tropics.

These gridded temperature changes are combined with the yield–temperature response function estimated using Eq. 1 and baseline growing-season temperature to give yield changes at different levels of global warming. We calculate yield changes for warming of 1, 2, and 3 degree Celsius including the estimated effect of on-farm adaptation. Any predicted yield losses >100% are set to losses of 99%. The $CO_2$ fertilization effect is included for all crops. $CO_2$ concentrations for a given level of global temperature change are determined based on a fitted quadratic relationship between global temperature change and $CO_2$ concentrations from the RCP 8.5 CMIP5 multi-model ensemble mean (adjusted $R^2 > 0.999$, 98 degrees of freedom).

**AgMIP GGCMI ensemble**. The AgMIP GGCMI is the one other source of global, multi-crop, multi-model yield responses and so we compare the results of our meta-analysis against these results. This ensemble of gridded crop model outputs includes up to seven process-based crop models, run using five General Circulation Models (GCMs)[18]. Yield changes are calculated relative to the 1981–2000 average. In order to determine yield changes for specific levels of temperature change (1–3 °C), we find the year in which warming passes each specific level for each GCM for the RCP 8.5 emissions scenario (taking the average of multiple ensemble members, if available) and take the 11-year yield average around that year[40]. We determine irrigated areas using crop-specific irrigation areas from Monfreda et al[37]. and use irrigated results for cells where irrigated crop area exceeds non-irrigated crop area. We use runs including $CO_2$ fertilization for all analyses.

The results reported in the main text use a preferred AgMIP ensemble that only includes models that explicitly represent nitrogen stress (EPIC, GEPIC, PEGASUS, and pDSSAT). We believe these results are preferred given crop response to changing temperature and, in particular, $CO_2$ conditions is known to depend on nutrient availability[41,42] and crops in many areas of the world are currently under-fertilized[43]. Moreover, the distinction between models based on representation of nitrogen stress has been identified as significant in understanding ensemble results by the AgMIP team[18]. Results in the main text should therefore be interpreted as impacts assuming continuation of current nutrient management practices. In the SI, we report results using the full AgMIP ensemble, which differ substantially from those of the restricted ensemble (Supplementary Fig. 11 and Supplementary Table 5). For both ensembles, the mean is calculated as a simple mean of yield change for each level of warming using all crop model×GCM combinations.

**Welfare consequences of yield changes**. To estimate the economic implications of warming-induced yield shocks, we use the Global Trade Analysis Project (GTAP) general equilibrium model and its accompanying database[23,24]. GTAP is a widely used, comparative static general equilibrium model that exhaustively tracks bilateral trade flows between all countries in the world and explicitly models the consumption and production for all commodities of each national economy. Producers are assumed to maximize profits, while consumers maximize utility. Factor market clearing requires that supply equal demand for agricultural and non-agricultural skilled and unskilled labor and capital, natural resources, and agricultural land, and adjustments in each of these markets in response to the climate change shocks determines the resulting wage and rental rate impacts. The model has been validated with respect to its performance in predicting the price impacts of exogenous supply side shocks, such as those that might result from global climate change[44]. Additional information on the structure of GTAP is given in Supplementary Fig. 22.

GTAP captures a number of dimensions important for determining the welfare implications of climate change impacts on agriculture. These include the shifting of land area between crops, potential intensification of production, shifting of consumption between commodities and sources of goods, and the adjustment of global trade patterns (Supplementary Table 2). For the purposes of this study, GTAP is run with 140 regions and 14 commodities—with the latter designed to place an emphasis on the agricultural sector. Productivity changes are introduced to GTAP as a Hicks-neutral shift in the production function such that farmers employing the same combination of inputs would experience $X$% lower output in the presence of a $X$% climate-driven yield shock.

Wheat and rice are modeled as individual sectors within each region. Maize is part of the coarse grains sector and soybeans is part of the oilseeds sector. Impacts in these sectors are scaled downwards based on the relative importance of maize and soybeans for sectoral production in each region. Yields of crops not covered in the meta-analysis (coarse grains nec., oilseeds nec., sugarcane, cotton, and fruits and vegetables) are not altered. Absent normalization, this will lead to an underestimate of potential climate impacts, since these other sectors are also likely to be affected by climate change. Therefore, in the results that follow, welfare changes are normalized by the value of production of the crops covered in the meta-analysis. Global and regional welfare changes are measured in terms of equivalent variation and are decomposed into the three components shown in Fig. 2 following Hertel and Randhir[25].

In order to explore uncertainty in the economic modeling, we perform a systematic sensitivity analysis of GTAP output to perturbations in four sets of key parameters governing the supply and demand behavior in this model. On the supply side, these pertain to the parameters determining the intensive (substitution of other inputs for land) and extensive (land supply elasticities) margins of crop supply response to commodity price. On the demand side, these are the parameters that govern the price elasticity of demand for food and the price elasticity of demand for imports—which in turn govern the price responsiveness of export demands. Parameters vary by commodity/sector. We develop symmetric, triangular distributions for each parameter value, based on estimates in the literature (Supplementary Table 8) and sample from these distributions using the Gaussian Quadrature approach implemented by Arndt[45]. This approach has been shown to perform nearly as well as a complete Monte Carlo analysis in the context of CGE modeling, but it is much more efficient, requiring far fewer model solutions[46]. Due to the computational burden of conducting a complete, systematic sensitivity analysis in the 140 region model, we collapse those regions down to the 16 FUND regions for purposes of this robustness check. The resulting mean and standard deviations for regional welfare are reported in Supplementary Fig. 13. Because on the dominance of direct effects (i.e., the impact of climate change in yields) in many of the regions' total welfare, variation of the economic parameters has a modest impact on the underlying uncertainty.

**Calculating the social cost of carbon**. Results of the economic modeling are used to create damage functions that relate changes in economic welfare (measured as percentage of the value of agricultural sector output) with temperature change. GTAP results are aggregated from the country level to the 16 FUND regions. Damage functions are based on a linear interpolation between the point-estimates of welfare changes at 1, 2, and 3 °C of warming and then a linear extrapolation beyond 3 °C (results reported in main text) or on a quadratic fitted through the point estimates (Supplementary Table 5).

These agricultural damage functions are then incorporated into a sectorally and regionally disaggregated SCC damage module based on the FUND model, keeping the rest of the impact sectors unchanged[28, 47]. Damage functions in the module use the central parameter estimates of FUND. The full FUND model includes probability distributions over many parameters and is designed to be run in a Monte Carlo mode[47]. This uncertainty is not dealt with in this paper, meaning uncertainty reported in the SCC reflects only the uncertainty in the yield response derived from the meta-analysis. The damage module is driven by a standardized socio-economic and emissions pathway and climate model[28]. We use a business-as-usual emissions scenario (Scenario 2 in ref. 3), paired with the DICE climate module[48]. This produces a warming of 4 °C of warming above preindustrial by 2100. The SCC is calculated by adding a 1 Gt pulse of $CO_2$ emissions to this reference emissions path in 2020 and comparing the time path of damages along the perturbed pathway to the reference case. Then these incremental damages (or benefits) are discounted back to 2020 at a 3% discount rate and normalized by the $CO_2$ pulse volume to give the SCC. Results using alternative discount rates are given in Supplementary Table 5. As the SCC is additive, it can be decomposed by sector and region, allowing a detailed comparison of the regional impacts in agriculture between FUND and the revised regional damage functions.

**Code availability**. Code for GTAP is open source and available for download at http://www.gtap.agecon.purdue.edu/resources/res_display.asp?RecordID=2458. The FUND model is open source and available at http://www.fund-model.org/source-code. All other code is available from the authors upon request.

**Data availability**. Data for the meta-analysis is available at ag-impacts.org. Gridded yield changes based on the meta-analysis are available at 10.6084/m9.figshare.5417548. Welfare changes for GTAP regions based on yield shocks from both the meta-analysis and AgMIP are available at 10.6084/m9.figshare.5417557 and 10.6084/m9.figshare.5417560. Other data are available from the authors upon request.

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

## Acknowledgements

We thank Max Auffhammer, Sol Hsiang, Jerry Nelson, and David Lobell for comments on earlier drafts of this paper. T.H. acknowledges the support of the National Science Foundation (award 0951576) under the auspices of the RDCEP project at the University of Chicago and USDA-NIFA, Hatch Project 1003642. F.C.M. acknowledges support of this project by USDA NIFA (award 12225279).

## Author contributions

F.C.M. conceived the paper, conducted the yield meta-analysis, and wrote the paper. U.B. ran the GTAP model and analyzed welfare results. T.H. ran the GTAP model and wrote the paper. D.D. ran the FUND damage module and wrote the paper.

## Additional information

**Competing interests:** The authors declare no competing financial interests.

