## [Peer Review File · Nature Communications]

Reviewers' comments:

Reviewer #1 (Remarks to the Author):

Review of “New Science of Climate Change Impacts on Agriculture Implies Much Higher Social Cost of Carbon”

The paper presents an updated climate damage function for the agricultural sector, modifies the FUND model to use this new damage function and presents results that show how the Social Cost of Carbon changes in this modified version of FUND. The new agricultural damage function is constructed in three steps: first, the authors use a meta-analysis of yield loss estimates to construct a response function of crops yields to temperature. Second, they use the GTAP model to convert these yield loss estimates into economic loss estimates. And finally, they create new simple damage functions that are incorporated into the FUND model. As an alternative to their meta-analysis results, the authors also use results from the AgMIP project.

This is an important paper that should eventually be published. It fills an important gap in the literature and uses the right tools to do so. Having said that, it will require a major revision to address issues in the following areas: a) there is a large number of statements that are simply not supported by the analysis in the paper. It should not be difficult to address these concerns; simple reformulations should be enough. b) The paper has almost no robustness checks or sensitivity analysis to key assumptions. This is entirely inappropriate for a paper in an area riddled with uncertainties. I make a number of specific suggestions below, but I would strongly encourage the authors to identify areas of uncertainty that warrant sensitivity analyses themselves, without prompts from a reviewer. c) The description of methods is very sparse, making it impossible for a reader to really understand what the authors have done in key areas. I flag these areas below. d) I have a number of methodological comments.

Unsupported statements

- 1) The title is not supported by the analysis. First, the paper does not demonstrate whether the increase in the SCC is driven by the new yield loss estimates used in this study, or by the use of only one general equilibrium model to estimate economic losses. My understanding of the existing FUND impacts is that they are some average from a number of different CGE models, so it seems feasible that most of the difference in the new estimates is driven by the choice of the GTAP model. Second, while the increases might seem large in percentage terms, this is all off relatively small numbers, where one is bound to get relatively large percentage changes for any adjustment. In absolute terms the increase for one of the two specifications seems relatively modest: about \$4 per ton of CO₂.
- 2) P2L47-P3L52: The (strong) statements in this paragraph seem to be supported by citation 2 and 3. Both of these seem to be gray-publications that have not been peer reviewed. Citing non-peer reviewed publications as support for such strong statements is not appropriate for a publication in a peer-reviewed journal. The authors should either cite peer-reviewed sources to support something like this, or present their own argument to support these statements (which seems beyond the scope of this paper).
- 3) P3L54-55: This is another statement that reads more like an editorial. Either the authors should cite a source that explains *why* this is a problem, or they need to actually present an argument themselves.
- 4) P3L58-50: Nothing here supports the statement that there is a “lack of an empirical basis for damages in the models”. If I look at the documentation of these models, they all seem to cite

studies that describe the empirical basis of their damage functions. The studies cited in the next sentence all seem to argue that the empirical basis should be improved, but not that there is no empirical basis.

- 5) P3L62-63: If the authors think that there is “general agreement” that IAMs are inadequate for policy analysis, they should cite sources for such a statement or not make such statements.
- 6) P3L63-64: To claim that no previous study has attempted to incorporate current knowledge of climate impacts into IAMs seems wrong, presumably the authors of these models used current knowledge when they constructed these models?
- 7) P3L66-70: The authors need to cite something to support these statements.
- 8) P4L72-3: These “this is the first” statements should all be taken out. This paper is a parameter update study. That is important, but not a case of a “first” of anything. The statement is also way too broad, this study only covers agriculture, whereas this statement suggests a much broader scope.
- 9) P4L83-4: I would really encourage the authors to not waste valuable journal space by constantly trying to point out that they are the “first”, especially when that seems not warranted.
- 10) P4L84: Claims that this study is more traceable than existing IAM damage functions seem inappropriate. As far as I can see both FUND and DICE are open source and have plenty of documentation available. In contrast, I find the description of this study far from complete to really understand what the authors did. Finally, it is unclear to me whether the authors plan to release all data and computer code for the study in this paper. If they don’t plan on doing that, they should certainly not claim that this study is more traceable than previous studies.
- 11) P7L129-130: Looking at figure S7, I can’t follow the conclusion that the results from the AgMIP study and the meta-study look broadly similar (except for maize).
- 12) P9 footnote c: All speculations about models that have not been investigated in this study should be removed, they are not backed up by the analysis of this study.
- 13) P10L187: Don’t cite studies that have not been peer-reviewed as support for statements.
- 14) P10L192: Don’t cite unpublished studies.
- 15) P11L203-4: This major claim of the study is actually not supported by the analysis because it might well be the choice of the specific CGE model in this study that explains the difference in results compared to the existing SCC estimates.
- 16) P11L204: The paper also does not show anything for models other than FUND.
- 17) P11L206-8: The paper provides so little robustness analysis as it stands, that any claim about a robust finding is not warranted at this point.
- 18) P12L216-7: This is pure speculation. The authors need to either cite sources that back such claims up or provide a legal and political analysis if they want to include such statements (clearly beyond the scope of this paper).
- 19) P12L219-20: The language here seems to suggest that one can expect a similar result for other sectors. That is speculation and should be reformulated.
- 20) P12L220-222: Haven’t IAMs demonstrated that this can be done for more than two decades?
- 21) P12L222-223: I don’t understand what is new about “this approach”? Haven’t previous damage functions used essentially a similar approach, i.e. reflected best science from the literature? This sounds like a novelty claim that I don’t see substantiated.

- 22) P12L223: Unless the authors plan to make all the code and data used in this analysis available without restrictions, they should not make claims that these results can be easily updated in the future.

Methodological comments

- 23) Excluding the CO₂ fertilization effect for C4 crops seems really arbitrary. This is supposed to be a meta-study, and that means that one takes the results of the underlying studies as they are and doesn't cherry pick studies. It might be appropriate to show results without CO₂ fertilization for C4 crops as a sensitivity analyses in an appendix.
- 24) I understand the motivation for the choices about the adaptation intercept, but I would like to see much more sensitivity analyses on this topic. For example, if there are studies in the sample that handle adaptation correctly (i.e. at 0 warming they would show the same impact as a study without adaptation), the choice in this manuscript will mishandle such a study, so I would like to also see results that take a different approach as a robustness check. I also worry about the asymmetric way the estimates without and with adaptation are treated. For example, one robustness check could be this: split the sample into with and without adaptation, and then run a regression without an intercept term for both samples, thus using the same structural constraint that is used for damage functions (i.e. the response function must include the origin).
- 25) How does the meta-analysis handle multiple estimates from the same group? For example, a modeling group could have published something, and then published updated estimates a couple of years later. Should these two publications receive the same weight?
- 26) The paper should present results for a range of discount rates, a standard practice in climate economics.
- 27) Why is the paper using a constant discount rate? All three IAMs mentioned in this paper normally use a Ramsey discount rate, and that seems the widely accepted standard approach in the literature that should also be used here.
- 28) P13L246: Why not use some kind of average baseline growing season temperature of the two countries?
- 29) P15L283: Why was this functional form picked? Is this standard in the literature?
- 30) P15L293-295: Do local temperature changes really scale in a predictable way with global temperature? In particular, do all models produce the same scaling pattern? Aren't the authors ignoring uncertainty in this scaling pattern by using a model mean?
- 31) P16L306-8: What is the justification for this approach?
- 32) P16L323-5: I see nothing in the cited source that describes the LPJ-GUESS model as an outlier. The way "calibration" seems to be used in the original source also doesn't suggest to me that this model should be dropped. Dropping a model like this seems really arbitrary. The authors could present a sensitivity analysis without the LPJ-GUESS model in an online appendix, but the main results should not present results from the AgMIP study with arbitrary alterations to the AgMIP model set.
- 33) P18L363: Why don't the damage functions also relate CO₂ concentrations to damages? Isn't that the structure of the original FUND function?
- 34) P18L366: There needs to be a sensitivity analysis around the choice of how to extrapolate beyond 3C.

- 35) Why is the paper using so many choices that EPA made when computing the SCC? Specifically, discount rates, scenarios and climate module. None of these choices have been peer-reviewed, why not just use the original FUND model as it was designed by its creator?

Clarifications

- 36) Figure 1 and all other numbers: what is the baseline for the temperature change numbers? Relative to pre-industrial, current day temperatures or something else?
- 37) Figure 1: are these figures with or without adaptation?
- 38) Figure 1: For soy this figure shows yield losses larger than 100%. That seems to make no sense. How are these results handled in the subsequent analysis?
- 39) P7L144: It would be good to use a little less econ jargon here. Maybe at least say equivalent variation of income, or something like that.
- 40) P9L173: Which version of the FUND model was used?
- 41) P10L184: What does “damage module based on the FUND model” mean? Are the authors not using FUND for this study?
- 42) P10L188: The authors should mention here what discount rate is used for these numbers.
- 43) P10L190: Could these results include some kind of uncertainty indication? Talking about the SCC in terms of point estimates strikes me as odd.
- 44) Figure 3: These figures are really hard to read, especially given that the error bars in some regions use up all the space, so that one can hardly distinguish the central estimates.
- 45) P11L204-5: Don’t use the plural for “damage functions”, the paper only shows things for the FUND damage function.
- 46) P13L249: In general, I assume everything here is based on yearly averages? Would be good to mention this somewhere.
- 47) P17L348: Spelling.
- 48) P18L370: Why is 3 cited here?
- 49) P18L370: If the model is driven by multiple scenarios, how are they aggregated?
- 50) P18L371: What is the standardized climate module used here? I thought the interagency workgroup just picked one distribution for the climate sensitivity, but the text here suggests more standardization?
- 51) P19L372: Just above it says that multiple scenarios were used, this sentence suggests that one BAU scenario was used. The reference seems incorrect here, so it is not clear what scenario is used.
- 52) Figure 4: it is not clear to me how uncertainty is handled in the computation of the SCC. Is the error bar for the meta analysis just based on uncertainty about the agricultural impacts? That doesn’t seem to make sense to me, doesn’t FUND also have uncertainty about many other parameters?

Reviewer #2 (Remarks to the Author):

Thank you for the opportunity to review this paper. I read it with great interest – it approaches an important topic in a unique way, and the analysis appears to be very thorough. The paper notes the current troubling “state-of-the-art” within the community of Integrated Assessment Model (IAMs), used to calculate the social cost of carbon (SCC). It notes that the damages used within these models are not based on the most up to date science or evidence, and so incorporates another source of damage information for the agricultural sector, specifically for staple crops. The authors create a set of temperature-yield impact functions from this, and use these to calculate a general equilibrium adjusted welfare impact of temperature on agriculture globally. This is then incorporated into one of the standard IAMs used for SCC calculations, and the authors find that incorporating these alternate agricultural damages increases the social cost of carbon.

While I think the motivation behind the analysis is laudable and it is a unique paper, I am not sure that as it currently stands it reaches its own stated goals of transparency. This is a problem, since the authors are aiming for a contribution that highlights their end-to-end method above current approaches. I feel it replaces one damage function with another well, but in what should be the broader contribution it provides less guidance on how to identify, evaluate, and incorporate damage functions from up-to-date evidence transparently in climate economic modeling generally.

I have four main concerns, which I will deal with first in no particular order. Miscellaneous comments follow.

1. Criteria for evidence

For this analysis to be insightful beyond the very specific case presented here, there needs to be a very clear argument about what type and standard of evidence should be used to calibrate the damage functions in a climate economic model. I think that is missing. The authors use a meta-analysis from Chaliner et al. (2014), and all the strengths and weaknesses of that survey of the literature are then absorbed by the current analysis.

This meta-analysis, even in the original paper, doesn't provide us with much critical guidance on what criteria there are for inclusion. For the current paper: What are the criteria for inclusion of the point estimates? How many of them are extensively empirically calibrated or validated? The original article claims some are, but the level of validation appears to vary upon inspecting papers cited therein. How many purely empirical models are there? (Chaliner et al. states that only 4 studies have purely empirical estimates and there does appear to be some systematic differences between these and the process model estimates.) A point estimate from a crop model will use biophysical parameters about a location, but we are left with no sense of whether these models are doing a good job of actually replicating reality, or of their skill in their projections. Most importantly, we have no sense of the consistency across these studies, and it is here where a major contribution could be made – through a systematic effort to standardize the evidence across all of these estimates and to justify it broadly.

Quite a few of these estimates are also over 20 years old, and the criteria for inclusion seems to be just that they have been published. Chaliner et al. note that these estimates may not represent the most up to date science themselves. The number of estimates is also misleading. The top 6 studies cumulatively give about 700 of these point estimates! The consistency and methods across studies is also opaque. Chaliner et al. mention this in terms of adaptation (which I discuss below) but it is likely to be true for a number of different facets of these papers. What are the assumptions, parameters, and calibrations that went into the models used? Was any effort made either by the authors doing the meta-analysis or the current authors to standardize or at least catalog the differences? This could be crucial, not only for adaptation, but also for assumptions

about local representativeness (discussed below), credibility of management practices, etc.

An attempt is made to compare the current meta-analysis to the GGCM dataset as validation. And yet this merely compares a set of models to another set of models which themselves have not been extensively validated. To date, I know of no published paper which systematically compares the GGCM to observed data. One that I have been referred to by members of the AgMIP group is currently in submission (Muller et al, available here: <http://www.geosci-model-dev-discuss.net/gmd-2016-207/#discussion>), and finds that the GGCM appears to have a number of limitations when evaluated systematically for predictive skill.

All of this leads me to ask: what is this current evidence representing, and how do we evaluate it for quality? For the current analysis to be the contribution that it claims to be, it needs to be a lot more critical and precise about what counts as evidence. Is a damage function better because it has more data behind it, or should we require that it is closer to the truth? I think the authors need to provide much more detail and transparency on the methodologies behind the point estimates included.

2. Treatment of spatial heterogeneity in the analysis

The treatment of spatial heterogeneity within the analysis is not particularly transparent. The authors state, and one would expect, that differences across space matter. The approach used in the paper abstracts away from this concern. If I follow it correctly, the authors first gather location specific estimates from papers. As above, it is unclear how specifically tailored to differing locations each crop model estimate is. These are then collapsed into a single impact conditional on projected temperature changes, which explicitly disregards spatial heterogeneity. The fact that in observed data there is substantial differences in yields and marginal damages from temperature is then lost in the analysis. These global, non-spatially heterogeneous impact functions are then applied to spatially heterogeneous warming to give crop impacts that are heterogeneous across space. This heterogeneity in TEMPERATURE is then used to model the equilibrium adjustments under trade. Finally, these welfare impacts are then aggregated up to the regions of FUND, where FUND's own model will apply regionally heterogeneous growth and climate projections (though the former appear to be based on a linear scaling of a global growth projection that preserves the (presumably) year 2000 distribution of global income).

This would not be especially troubling were it not for the fact that the regional results are presented in the paper as meaningful, but tracing the exact source of those regional differences seems difficult and untransparent. For the spatial aspect of this to be included in the paper, I would expect a much more detailed treatment (along with sensitivity analysis) of where each transformation of the initial global impact coefficient occurs, and how much distortion arises from choosing a globally representative coefficient.

3. Adaptation as a blackbox

The way the paper is written, it appears to be dealing with adaptation. However, it deals with only some forms, often inconsistently, and doesn't track the impact of different adaptation assumptions through the analysis. What determines whether adaptations occur? Are all farmers rational optimizers and so automatically adjust their planting behavior immediately? Do we observe any of the same adaptations in real data? Are these adaptations comprehensive? It seems to be limited to just marginal changes in cropping behavior and not adaptations based on changing crops, exit from agriculture, general shifts in cropping patterns, etc. In fact, it seems there is much left out, and that calling this adaptation overstates what is being included in the analysis. This needs to be made much clearer in the early stages of the paper, since the impression is of more comprehensiveness than actually is included.

Does each of the studies deal with adaptation in the same way – is there a unified treatment of

adaptation or an adaptation scenario within the estimates? Reading through the discussion of adaptation in the meta-analysis of Chaliner et al, it appears not. They point out that, "only two categories [of adaptation behaviours] have more than 20 entries in the paired adaptation studies". This part of the exercise becomes problematic.

There's also a few layers of adaptation here: minute marginal adjustments treated inconsistently across agronomic models (and presumably included implicitly in the quantitatively distinct statistical estimates in Chaliner et al), adjustments of production through a GE calculation in GTAP, and then the adaptation that is embedded into FUND itself. As far as I can tell, none of these sources of adaptation is well-founded in actual data, so using the term in the paper results in a complex blackbox. The authors should be very clear about the types of adaptation being discussed throughout, whether any of these have been observed in actual data as occurring, and treat them all coherently and consistently (i.e., by looking at alternative cases using ONLY adaptations of a particularly type, not a single dummy in the main analysis that groups ALL of the limited set of adaptations together). Importantly, they should give a detailed description of the types of adaptation that are not being considered, as currently the broad term "adaptation" is used without clarification that this leaves out some of the most important proposed adaptive behaviors.

If it is possible to reanalyze the papers in the meta-analysis to be consistent across studies, then something more can be said in the paper. Failing being able to treat this in a consistent way across all estimates and all steps of the process (including alternate analyses tracking the contributions of each to the final SCC), I feel it would be better to be clear about the limitations in the analysis, state that this will deal only with exactly quantifiable and consistent forms of adaptation or with none at all, and have an un-adaptation adjusted estimate of damages and the SCC as the main result. In my opinion, that is a more transparent approach than what is currently employed.

4. Does the paper establish a broad and generalizable approach to updating climate damages?

The overall question that I am left with is: does the currently proposed methodology provide sufficient guidance for a much-needed overhaul of the way that we in the IAM community deal with damages? While interesting, I think it is hard to draw the conclusions needed to do so. The headline number on the SCC may provide an interesting outcome of the analysis, but it is of less importance than the potential methodological contribution of providing a set of standards and criteria for what evidence to incorporate into the SCC calculation. It is generally accepted in the community at the moment that guidance on what evidence to include is of more importance than adjusting damage functions with no general guidance. It is my opinion that re-orienting the paper to be this kind of contribution is a worthwhile if substantial exercise.

5. Miscellaneous:

a. The exact critique of the current IAM damages (that they are out of date) is perhaps not the correct one. While in FUND the literature is out of date, as in the other models, the problem is less with the recency of evidence used and more about the type of evidence used, and the standards for inclusion. This perhaps represents more of a cultural drift (between impacts, integrated modelling, and damage communities) than it does any intentional failings in the models. The statements about "20 year old evidence" are less relevant than the fact that the evidence is opaque and somewhat arbitrary.

b. I'm confused as to why the local-to-global warming relationship was used to downscale the mean global temperature. Is there any reason to not use the gridded, spatially heterogeneous output of the CMIP5 ensemble itself? If there are clear reasons for this, it should be made clear in the text. To understand the difference outcome here, it would be helpful to see the current method compared to the gridded model output.

c. For the derivation of the damage function confidence intervals, it is noted that the SEs are calculated to block bootstrapping at the study level. Does this account for all agronomic model interdependencies? Is there any model that produces estimates across studies? If so, that would seem to be a better level at which to resample. This concern is perhaps a result of the lack of transparency about the methodologies underlying the point estimates in the meta-analysis.

d. The number of included point estimates varies in the paper. Chalinor et al has 1700, and the authors note that this decreases when restricting to the 4 crops used herein, to 1010, but later states that the number is 1030. I suspect it is the latter, since the crop specific numbers on page 13 of the manuscript are consistent with that.

e. The authors note that they show systematic differences from the GGCM across the tropics and temperature regions, and then judge this to not be important when aggregated globally. While this is true, the fact that the current paper values regional impacts in welfare terms means that the difference here is of importance. This once again argues for more clarity on how they are tracking spatial heterogeneity throughout the analysis, and whether this reflects a real and meaningful heterogeneity of evidence-based impacts.

Reviewer #3 (Remarks to the Author):

I found the topic of the paper interesting. The paper is well written and addresses a relevant question. The authors analyze the sensitivity of the forecasted social cost of carbon to the damage function used to predict climate change effect on crop yields.

I think that several issues need to be addressed by the authors before publication;

- The authors state that the current damage functions used in integrated assessment models are out of date (for example on line 62), but this is barely documented by the authors. What is the origin of these damage functions? Are these functions the same in all IAM? Is it proven that these "old" functions are inaccurate compared to more recent damage functions ?
- The authors wrote that they "present a new meta-analysis" (75). But what is really new in their meta-analysis? The dataset was already published by Challinor et al. (2014).
- The authors mention that their damage functions are "up-to-date" (85). I am not fully convinced, especially for wheat. For this crop, the authors used 336 yield data. This is very small compared to the size of the dataset used by Wilcox and Makowski (2014) in their meta-analysis (Wilcox and Makowski. 2014. A meta-analysis of the predicted effects of climate change on wheat yields using simulation studies. *Field Crops Research* 156, 180–190). I advise the authors using this dataset for wheat as it includes more than 1000 wheat yield data.
- The authors did not explain why they produced a new yield response function instead of using the function already published by Challinor et al. (2014). For wheat, they could also directly use the meta-regression model published by Wilcox and Makowski (2014). Why producing a new meta-regression?
- In lines 108-112, the authors explain that the CO₂ function does not produce realistic values. I advise the authors using a different function. For example, they could replace their $f(\text{CO}_2)$ function by a quadratic function whose parameters could be directly fitted to the data. They could also try the meta-regression models mentioned above.
- The statistical model is presented on page 13, but the authors do not explain how this model was selected. This model should be compared to other models using standard statistical methods. In addition, the authors should provide the readers with all parameter values. Only three parameter values are presented in Table S1. Why? Finally, the authors should analyze the distribution of model residuals in order to check whether the assumptions made by their model are realistic.
- In eq.(1), the effects of some variables are crop-dependent, while the effects of others are the same for all crops. Why?

Response to Reviewers

We thank the reviewers for their detailed and constructive feedback. The manuscript has been substantially revised (and improved) in response to comments. Major changes to the manuscript include:

- 1) Inclusion of a CO₂ fertilization response for C₄ crops
- 2) Different treatment of the AgMIP GGCM ensemble, including a sensitivity analysis around the set of models to include (all models vs models that explicitly represent nitrogen stress)
- 3) A systematic sensitivity analysis of economic modeling parameters in GTAP
- 4) Additional robustness checks for the yield response meta-analysis including type of study (empirical vs process-based), date of study, and type of adaptation included
- 5) Additional robustness checks for the SCC using alternative discount rates and an alternative method for extrapolating beyond 3 degrees

We do not find these changes alter the conclusions in the original manuscript. Below we provide a point-by-point response to reviewer comments.

Reviewer 1

Unsupported statements

1) The title is not supported by the analysis. First, the paper does not demonstrate whether the increase in the SCC is driven by the new yield loss estimates used in this study, or by the use of only one general equilibrium model to estimate economic losses. My understanding of the existing FUND impacts is that they are some average from a number of different CGE models, so it seems feasible that most of the difference in the new estimates is driven by the choice of the GTAP model. Second, while the increases might seem large in percentage terms, this is all off relatively small numbers, where one is bound to get relatively large percentage changes for any adjustment. In absolute terms the increase for one of the two specifications seems relatively modest: about \$4 per ton of CO₂.

- The title has been changed from “New Science of Climate Change Impacts on Agriculture Imply Much Higher Social Cost of Carbon” to “New Science of Climate Change Impacts on Agriculture Imply Higher Social Cost of Carbon”
- - The reviewer notes that six different CGE models have been used to develop the current FUND damage function. However, we find that several of these models rely on the same GTAP data base used in this study (described more fully below). This data base also includes the key behavioral parameters used in this study, although some of the models alter these in their own work. In short, when it comes to the underlying data and model structure, many global CGE models currently used in climate impact analysis share the same data and basic structure as the standard GTAP model used here. More specifically:
 - o The economic models upon which the agricultural damages in FUND are built were constructed in the 1980’s and early 1990’s. Three of the estimates are derived from publications written by scientists who, at the time, were at ERS/USDA. The first two of

these (Reilly *et al* 1994, Kane *et al* 1992) were executed using a partial equilibrium model of commodity trade, nicknamed SWOPSIM. This was developed at ERS/USDA in the 1980's and was widely used for the analysis of commodity trade policy. As such, this model did not explicitly identify the factors of production (land, labor, capital, materials) underlying crop production. This limited their effectiveness for analysis of climate change. The model also only considered net trade flows, ignoring the unique, bilateral relationships which characterize international agricultural trade. As a consequence, when the first GTAP data base and model became available, ERS/USDA hired one of the co-authors of that model (Tsigas) and used a variants of the GTAP model for subsequent climate impact assessments (Tsigas *et al* 1996). The model which they employ in the 1996 publication has the same basic structure as the one used here. However, the parameters, data and policies have subsequently been updated 8 times (we use GTAP v.9 here).

- ...
- ○ The next line of work is from NIES in Japan (Morita *et al* 1994). This model, nicknamed AIM, is an integrated assessment model which, over time, has adopted the GTAP data base and model structure for its agricultural module. As such, it has embraced the model structure, data and parameters which form the basis for the central case analysis presented in this paper. [Note that Morita et al (1994) is cited as a reference for agricultural damages in FUND documentation (Anthoff and Tol 2014) but is not cited in either of the supporting papers (Tol 2002a, 2002b)].
- ○ The final strand of work in this area is based on the research of Klaus Frohberg and Guenther Fischer who developed this at IIASA (Fischer *et al* 1996). This framework utilized a system of linked national models. In this case, there is no common model structure, but rather individual country authors design their own system. While conceptually attractive, in practice this makes it extremely difficult to understand model results and virtually impossible to undertake systematic sensitivity analyses such as the one offered in our revised paper (see below).
- - Recognizing that the CGE model itself is a potential source of uncertainty in linking the SCC to climate-induced yield changes, we have undertaken extensive robustness checks to investigate the extent of this uncertainty. Because of the widespread use of the GTAP database and parameter estimates in climate impact modeling, we believe that the most useful type of sensitivity analysis involves an extensive exploration of the GTAP parameter space. We have now done so, using the Gaussian Quadrature approach which is computationally efficient, but which gives similar results to Monte Carlo analysis (DeVuyst and Preckel 1997). Parameter distributions are given in Table S7. We find that the regional welfare results are quite robust to this variation (Figure S13). This is not surprising when one refers to the decomposition of these sources of welfare change. The largest share comes from the direct effect of the productivity changes. These are not dependent on the behavioral parameters. Rather it reflects the interaction between the value of crop production and the productivity shocks derived from the meta-analysis (Huff and Hertel 1996).
- - We note that results from the GTAP model used in previous FUND calibrations (Tsigas *et al* 1996) do not appear to be an outlier among the set of references used in the meta-analysis (Table 1, Tol 2002b). Therefore it is unlikely that the choice of CGE model would drive the differences in results we document.

- We also note that our results demonstrate a clear sensitivity to climate change agricultural productivity impacts, as evidenced by the difference in SCC calculated using the meta-analysis and the two AgMIP ensembles (Table S4). Nevertheless, all imply a higher SCC than currently given in FUND. This makes it unlikely that our conclusions are an artefact of the chosen CGE model.

2) P2L47-P3L52: The (strong) statements in this paragraph seem to be supported by citation 2 and 3. Both of these seem to be gray-publications that have not been peer reviewed. Citing non-peer reviewed publications as support for such strong statements is not appropriate for a publication in a peer-reviewed journal. The authors should either cite peer-reviewed sources to support something like this, or present their own argument to support these statements (which seems beyond the scope of this paper).

- The reference for this statement has been changed to a recent report by the National Academy of Sciences on the social cost of carbon, which is peer reviewed and contains the same information as the EPRI report initially cited regarding documentation of the damage functions.

3) P3L54-55: This is another statement that reads more like an editorial. Either the authors should cite a source that explains why this is a problem, or they need to actually present an argument themselves.

- This paragraph has been changed to: “The lack of a current empirical basis for IAM damage functions is not just an academic question because the SCC has been formally adopted by the U.S. government to quantify the benefits of CO2 mitigation in cost-benefit analysis (IAWG 2013). Regulations with benefits totaling over \$1 trillion have used the SCC in cost-benefit analysis.(Nordhaus n.d.) Increasingly it is also being used at the state level: recent rulings in California, New York and Minnesota all require use of the SCC in analysis of climate and energy regulations (Schlatter 2016, California 2016, Larson 2016). Therefore, the value of the SCC is a relevant consideration for long-term planning in industry and government, and yet the extent to which it would change if more recent scientific knowledge were incorporated is largely unknown. Many recent commentaries on the state of climate change economics have identified improving the empirical basis of IAM damage functions as high-priority area for future work (Burke *et al* 2016, Pindyck 2013, Revesz *et al* 2014, Stern 2016, Carleton and Hsiang 2016, NAS 2017).”

4) P3L58-50: Nothing here supports the statement that there is a “lack of an empirical basis for damages in the models”. If I look at the documentation of these models, they all seem to cite studies that describe the empirical basis of their damage functions. The studies cited in the next sentence all seem to argue that the empirical basis should be improved, but not that there is no empirical basis.

- This statement has been removed

5) P3L62-63: If the authors think that there is “general agreement” that IAMs are inadequate for policy analysis, they should cite sources for such a statement or not make such statements.

- This statement has been removed

6) P3L63-64: To claim that no previous study has attempted to incorporate current knowledge of climate impacts into IAMs seems wrong, presumably the authors of these models used current knowledge when they constructed these models?

- This statement has been removed

7) P3L66-70: The authors need to cite something to support these statements.

- This statement has been removed and the paragraph changed to the following:

“A large and growing body of science has dramatically improved our knowledge of the social and economic risks posed by climate change in recent years, and therefore provides an opportunity to substantially improve the empirical basis of the damages underlying the SCC (IPCC 2014, Carleton and Hsiang 2016). However, translating the literature on biophysical climate change impacts into damage functions is not straightforward. For each sector, the available science must be aggregated and translated into a consistent set of global impacts. Then, because damage functions parameterize how economic welfare changes with temperature, the welfare consequences of these biophysical impacts must be assessed. Finally new damage functions must be introduced into an IAM in order to examine the effect on the SCC. The complexity and interdisciplinary nature of this process may be part of the reason damage functions in IAMs have lagged the science of climate change impacts.”

8) P4L72-3: These “this is the first” statements should all be taken out. This paper is a parameter update study. That is important, but not a case of a “first” of anything. The statement is also way too broad, this study only covers agriculture, whereas this statement suggests a much broader scope.

- This statement has been removed and replaced with “This paper does this for the agricultural sector, connecting a current and comprehensive review of the biophysical science of impacts of climate change on yields to the SCC.”

9) P4L83-4: I would really encourage the authors to not waste valuable journal space by constantly trying to point out that they are the “first”, especially when that seems not warranted.

- This statement has been removed and replaced with “This is therefore an “end-to-end” analysis that directly connects the biophysical science of climate change impacts to the SCC in a single study.”

10) P4L84: Claims that this study is more traceable than existing IAM damage functions seem inappropriate. As far as I can see both FUND and DICE are open source and have plenty of documentation available. In contrast, I find the description of this study far from complete to really understand what the authors did. Finally, it is unclear to me whether the authors plan to release all data and computer code for the study in this paper. If they don’t plan on doing that, they should certainly not claim that this study is more traceable than previous studies.

- The statement about traceability here refers specifically to the connection between parameterization of the damage function and the underlying scientific literature on climate change impacts, rather than the documentation of equations in IAMs, which is generally good. While IAM documentation may include citations to supporting literature, the traceability from those citations to parameterization of the damage function is often difficult or impossible to follow. The most relevant example is the FUND agriculture sector damage function. The impacts of changing temperature on agriculture in FUND are given by quadratic damage functions for each region. The reference for this is a meta-analysis of five studies in Tol (2002b). However, this reference provides a point estimate for impacts at 2.5 degrees, which is insufficient to calibrate a quadratic. The calibration of the second data point for the quadratic is in fact documented in a different paper, Tol (2002a). Moreover, both these papers describe calibration of damages for a

9 region version of FUND. Calibration of damages for the current 16 region version of FUND are undocumented. Finally, Morita et al. (1994) is cited as a reference in the FUND documentation but is not cited in either Tol (Tol 2002a) or Tol (Tol 2002b).

- This statement has been modified to read “This means our damage functions have a clear and traceable connection to the underlying science that is both comprehensive and up-to-date, in contrast to most current IAM damage functions.”
- We will make the code, supporting data, and data-products fully available to the community. The code for the meta-analysis and resulting gridded-yield files (as well as regional welfare results from GTAP) will be made available on Frances Moore’s website. The yield impacts database used for the meta-analysis is available at www.ag-impacts.org. AgMIP data is already publicly available as part of the ISIMIP Fast Trak project. GTAP code is publically available and open source. The specific GTAP model used for this analysis will be made available and can be run by anyone with a license to the GTAP v9 database.

11) P7L129-130: Looking at figure S7, I can’t follow the conclusion that the results from the AgMIP study and the meta-study look broadly similar (except for maize).

- This statement has been changed to the following: “At the global scale, these cancel to some degree so that differences in global, production-weighted yield changes between the two methods are smaller and, due to large uncertainties involved, statistically-indistinguishable from each other (Figure S7).”

12) P9 footnote c: All speculations about models that have not been investigated in this study should be removed, they are not backed up by the analysis of this study.

- This footnote has been removed.

13) P10L187: Don’t cite studies that have not been peer-reviewed as support for statements.

- These references have been removed

14) P10L192: Don’t cite unpublished studies.

- We believe this information is important for readers to understand our results within the overall context of damages in FUND. As far as we know, the decomposition of total damages by sector has only been undertaken in the cited studies. Nature Communications does not appear to have an editorial policy regarding not citing technical reports. We have removed the citation to Diaz (2014), but retained the reference to the Rose et al. (2014) (which is published, but not peer-reviewed). Since the report is public and freely available, readers are easily able to form their own judgments about the quality of the findings.

15) P11L203-4: This major claim of the study is actually not supported by the analysis because it might well be the choice of the specific CGE model in this study that explains the difference in results compared to the existing SCC estimates.

- See response to comment 1

16) P11L204: The paper also does not show anything for models other than FUND.

- This statement has been clarified to “Here we show that the current science of climate change impacts on agriculture implies far larger damages to the sector than currently represented in models used to calculate the SCC.”
- Since only FUND represents impacts in the agricultural sector, we believe this statement is accurate.

17) P11L206-8: The paper provides so little robustness analysis as it stands, that any claim about a robust finding is not warranted at this point.

- This statement has been modified to make it clear the claim of robustness only applies to uncertainties investigated in the paper, including additional robustness checks added in response to reviewers’ comments:

“Though uncertainty in the impacts of climate change on yields is substantial, our finding that the SCC should be increased is robust to this uncertainty, as well as to uncertainties relating to the discount rate, economic modeling, and extrapolation of the damage function.”

18) P12L216-7: This is pure speculation. The authors need to either cite sources that back such claims up or provide a legal and political analysis if they want to include such statements (clearly beyond the scope of this paper).

- The statement has been changed to: “Governments are currently relying on IAMs to evaluate climate and energy policy and these models have already come under legal scrutiny as a result.(MPUC 2016, Appeals 2016) It is therefore important, from both a regulatory and an academic perspective, that the representation of damages reflect current scientific consensus on impacts in a timely and transparent manner.” The two references are to legal opinions that have already been delivered in response to legal cases brought around the SCC.

19) P12L219-20: The language here seems to suggest that one can expect a similar result for other sectors. That is speculation and should be reformulated.

- We do not believe this is what is conveyed by this sentence. Rather the language reflects the fact that it may be surprising for readers not familiar with the representation of damages in the FUND model that changing a single sector in a model with 14 damage sectors leads to such a large proportional change in the total SCC. We have changed the language to “Here we have shown that improving the empirical basis of just the agricultural sector results in a large increase in the SCC.”

20) P12L220-222: Haven’t IAMs demonstrated that this can be done for more than two decades?

- The IAMs discussed here (i.e. DICE, PAGE, and FUND) do not represent the biophysical impacts of climate change directly, rather they parameterize the results of economic studies of climate change effects on welfare to produce a damage function. This means that the damage functions are at best several publications removed from the underlying science, making the empirical evidence that supports their parameterization difficult if not impossible to trace. The FUND documentation, for example, cites Tol (2002b) for the parameterization of the agricultural damage functions, which in turn cites 5 economic studies of the welfare effects of climate change impacts on agriculture. These in turn are based on parameterizations of the underlying biophysical impacts. For instance Kane (1992) references Ritchie et al. (1990), Rosenzweig (1990), Peart et al. (1990), and Santer (1985) as their basis for the biophysical impacts of climate change on yields. The novelty of this study is in integration – in incorporating across disciplines in order to integrate all aspects of this analysis. This is an important step

forward in reconciling assumptions across different modeling steps, and in reducing the lead time from scientific results to incorporation into IAM damage functions.

- Other kinds of IAMs (e.g. GCAM, IMAGE etc) do (or could) represent the biophysical impacts of climate change, but these have not historically produced a social cost of carbon.

21) P12L222-223: I don't understand what is new about "this approach"? Haven't previous damage functions used essentially a similar approach, i.e. reflected best science from the literature? This sounds like a novelty claim that I don't see substantiated.

- See response to previous comment.

- The sentence has been modified to clarify what we believe to be the novelty of the approach: "This approach, integrating analysis of biophysical climate change impacts, economic modeling of welfare effects, and calculation of the SCC, therefore represents an essential step in maintaining and improving the integrity of IAM results going forward."

Methodological comments

23) Excluding the CO₂ fertilization effect for C₄ crops seems really arbitrary. This is supposed to be a meta-study, and that means that one takes the results of the underlying studies as they are and doesn't cherry pick studies. It might be appropriate to show results without CO₂ fertilization for C₄ crops as a sensitivity analyses in an appendix.

- The methods have been modified to include a CO₂ fertilization effect for maize in the meta-analysis. The effect is allowed to differ from that of C₃ crops, but is of similar magnitude. This somewhat improves yields and reduces the negative welfare effects, but does not qualitatively change the results. All results in the current version of the paper include the CO₂ fertilization effect for maize.

24) I understand the motivation for the choices about the adaptation intercept, but I would like to see much more sensitivity analyses on this topic. For example, if there are studies in the sample that handle adaptation correctly (i.e. at 0 warming they would show the same impact as a study without adaptation), the choice in this manuscript will mishandle such a study, so I would like to also see results that take a different approach as a robustness check. I also worry about the asymmetric way the estimates without and with adaptation are treated. For example, one robustness check could be this: split the sample into with and without adaptation, and then run a regression without an intercept term for both samples, thus using the same structural constraint that is used for damage functions (i.e. the response function must include the origin).

- The meta-analysis is calculating the average effect of adaptation across all the studies. If one study were treating adaptation correctly, this would reduce the magnitude of the intercept term – it wouldn't be treated incorrectly in the analysis. We do not believe the proposed robustness check is desirable because 1) splitting the sample does not change the fact that adaptation is treated incorrectly in studies that purport to include adaptation, and that therefore not including an intercept term for the adaptation studies in the split sample would produce the same erroneous conclusion as to the benefits of adaptation as not including the same term in the full sample, and 2) splitting the sample changes the set of observations for non-adaptation variables (e.g. temperature response, CO₂ etc) that could introduce both bias and variance into these estimates

- Instead we investigated including an intercept term in the full regression. This allows studies that don't include adaptation to not pass through the origin and means that studies with and without adaptation are treated in the same way (i.e. by allowing an intercept term and subtracting this intercept in estimating the effect of temperature). The intercept term was found to be 1.44% with a standard deviation of 3.7%. It is substantially smaller than the intercept for the adaptation studies, of 6.30%. This alteration did not change the temperature response.
- In addition, in response to comment 3 from Reviewer 2, we investigated different types of adaptations, specifically adjusting planting date and cultivar, by far the most common types of adaptation reported in the database. Rather than including a single adaptation intercept and temperature interaction, we include two – one for studies that include changing planting date, and one for studies that include changing cultivar. The results are documented in Table S6. Thought point estimates do differ between adaptation types, standard errors are large and we cannot reject the hypothesis either that they are equal to zero or equal to the common effect estimated in our preferred model.

25) How does the meta-analysis handle multiple estimates from the same group? For example, a modeling group could have published something, and then published updated estimates a couple of years later. Should these two publications receive the same weight?

- The meta-analysis does assume independence between different studies, which may be invalid if there are correlations in the residuals between papers from the same authors or between papers using the same crop model. We have added more detail on this assumption and the set of models used to the Methods section: "This treatment of the errors does assume independence between studies, which may be questionable if the same model is used in multiple studies. In total, 28 models are represented in the database, made up of 17 process-based model families¹ and 11 statistical models, are used in the 52 studies."

26) The paper should present results for a range of discount rates, a standard practice in climate economics

- SCC has been calculated using 2.5%, 3%, and 5% discount rates. The discount rate does not qualitatively change the findings of the study. A table showing how the SCC varies in response to multiple sensitivity checks (yield models, discount rates, extrapolation of the damage function) has also been added to the supplemental information (Table S4).

27) Why is the paper using a constant discount rate? All three IAMs mentioned in this paper normally use a Ramsey discount rate, and that seems the widely accepted standard approach in the literature that should also be used here.

- In using a constant discount rate, we follow the approach used by the Inter-Agency Working Group that calculated the SCC used by the US federal government. The Office of Management and Budget also recommends constant discount rates be used in cost-benefit analysis.

28) P13L246: Why not use some kind of average baseline growing season temperature of the two countries?

¹ i.e. treating CERES-maize, CERES-rice and CERES-wheat as a single model

- Given the relatively small effect of baseline temperature on yield impacts, and given that most studies are looking at geographically contiguous countries with very similar growing-season temperatures, this is highly unlikely to be a major source of variation.

29) P15L283: Why was this functional form picked? Is this standard in the literature?

- In selecting this functional form, we considered two alternatives, both found to be inferior to the chosen form. A quadratic form is inconsistent with theory because it shows a negative marginal effect of CO₂ at high temperatures. This is because most of the data falls around 560ppm when warming at 3-5 degrees results much higher concentrations, which fall past the optimum of the estimated effect.
- We also investigated a logarithmic response function. This is not preferred because there was found to be a very strong dependence on the point chosen as baseline CO₂, related to the behavior of the logarithmic function close to zero.
- The chosen functional form allows flexibility over both the saturation point (fit through the regression) and the rate at which the saturation point is reached (through the “A” parameter”).
- There is not really a standard in the literature, because parameterizing CO₂ impact on yields in this way is unusual. Empirical yield studies do not include the CO₂ effect whereas process-based crop models do not parameterize the CO₂ effect. Rather it emerges from other functional forms describing plant water stress, nutrient availability, harvest index etc. The original Chalinor et al. (2014) meta-analysis used a linear response to changing CO₂, but we believe this to be inappropriate for extrapolating out to large CO₂ changes associated with 2-3 degrees of warming as is done in this study.

30) P15L293-295: Do local temperature changes really scale in a predictable way with global temperature? In particular, do all models produce the same scaling pattern? Aren't the authors ignoring uncertainty in this scaling pattern by using a model mean?

- Most pattern scaling driven by well-understood physical laws captured in all climate models such as the albedo feedback at high latitudes and the relative heat capacity of land vs ocean. These tend to be well-represented in climate models leading to relatively little spread between models in the pattern of warming. For instance Knutti and Sedlacek (2012) find that the spatial pattern of warming in the CMIP5 ensemble shows “good agreement” to “very good agreement” between models (based on a robustness measure based derived from inter-model spread, climate signal, and weather noise). They also show very strong similarities in the multi-model mean temperature scaling pattern between CMIP3 and CMIP5, indicating robustness across model generations that differ in model processes and spatial resolution.

31) P16L306-8: What is the justification for this approach?

- The figure below shows CO₂ concentration and temperature combinations for RCP 8.5 CMIP5 multi-model ensemble mean (green diamonds) with the fitted values based on a quadratic relationship shown in the blue line. The fitted model with 98 degrees of freedom has an adjusted R² of over 0.999.

- This information on model fit has been added to the main text: “CO₂ concentrations for a given level of global temperature change are determined based on a fitted quadratic between global temperature change and CO₂ concentrations from the RCP 8.5 CMIP5 multi-model ensemble mean relationship (Adjusted R² >0.999, 98 degrees of freedom).”

32) P16L323-5: I see nothing in the cited source that describes the LPJ-GUESS model as an outlier. The way “calibration” seems to be used in the original source also doesn’t suggest to me that this model should be dropped. Dropping a model like this seems really arbitrary. The authors could present a sensitivity analysis without the LPJ-GUESS model in an online appendix, but the main results should not present results from the AgMIP study with arbitrary alterations to the AgMIP model set.

- The results based on the AgMIP GGCMs have been substantially modified to address the reviewers’ concerns. We now present two results based on the AgMIP ensemble: our preferred results only include the set of models that include representation of explicit nitrogen stress (EPIC, GEPIC, pDSSAT, and PEGASUS). In the supplementary information we also present results using the full AgMIP ensemble. This difference does have a major effect on the results – the SCC increases from baseline FUND estimates using the full ensemble, but only by a small amount. Using the restricted ensemble, results are intermediate between the FUND and the meta-analysis results. We note this sensitivity at several points throughout the text:
 “The largest sensitivity of results is around including the full ensemble of process-based crop models in the AgMIP GGCMs. Adding models that don’t represent nitrogen stress substantially reduces the estimated impact of climate change on agriculture and leads to an SCC (\$9.6 per ton with a 3% discount rate) only slightly greater than the current FUND estimate. This difference is due to large productivity gains in many parts of the world (Figure S11a), a product particularly of the LPJ-GUESS and GAEZ-IMAGE models.(Rosenzweig *et al* 2014)” Results Section (p. 12)
 “In the supplementary information we report results using the full AgMIP ensemble, which differ substantially from those of the restricted ensemble.” Methods Section, (p. 19)
- We believe the ensemble of models that represent nitrogen stress is strongly preferred agronomically to the full ensemble. Crop response to changing temperature and (in particular) CO₂ conditions is known to depend on nutrient availability (for instance Leakey *et al* 2009, Reich

et al 2014) and crops in many areas of the world are under-fertilized (Mueller *et al* 2012). This distinction between models that do and do not represent nitrogen stress is one highlighted as important by the agronomists that created the GGCM (Rosenzweig *et al* 2014). Moreover, two of the three models excluded in the restricted ensemble (GAEZ-IMAGE and LPJ-GUESS) represent potential yields rather than actual yields and are therefore uncalibrated to observations (Rosenzweig *et al* 2014).

33) P18L363: Why don't the damage functions also relate CO₂ concentrations to damages? Isn't that the structure of the original FUND function?

- The FUND agricultural sector includes separate damage functions for warming, CO₂ concentration and adaption. This study instead provides a single agricultural sector damage function. Given the close connection between CO₂ and temperature, we felt there was little to be gained from separating out the two effects (the one exception would be an investigation of the benefits of solar radiation management, which separates the CO₂ effect from temperature). In addition, it is not clear that the welfare effects of CO₂ and temperature are additively separable, as currently assumed in the FUND representation of agricultural impacts.

34) P18L366: There needs to be a sensitivity analysis around the choice of how to extrapolate beyond 3C.

- We have added a second method for extrapolating beyond 3°C of warming. A quadratic was fitted through the GTAP results in each region and the fitted values used as the resulting damage function, including extrapolating beyond 3 degrees. These results are documented in Table S4. The alternative extrapolation using a fitted quadratic slightly increases the SCC but does not alter any conclusions of the paper.

35) Why is the paper using so many choices that EPA made when computing the SCC? Specifically, discount rates, scenarios and climate module. None of these choices have been peer-reviewed, why not just use the original FUND model as it was designed by its creator?

- Using a standardized socio-economics and climate scenario allows comparison across the three models used by the inter-agency working group and isolation of the difference in the SCC due to the representation of climate damages. Although in the initial submission we only referenced damages in FUND, in the revised manuscript we have added comparisons of the SCC just from our agricultural damages to the SCC from a DICE damage module and from the PAGE market damages. Although neither model explicitly represents the agricultural sector, agricultural impacts are included within both sets of damages, providing important context for the results from this new agricultural damage function. Using the standardized socio-economics and climate modules allows the differences between the SCC to be interpreted only as differences in representation of damages, making this comparison meaningful.

Clarifications

36) Figure 1 and all other numbers: what is the baseline for the temperature change numbers? Relative to pre-industrial, current day temperatures or something else?

- Temperature changes are relative to the 1995-2005 average, the median date of studies included in the meta-analysis. This has been clarified in the main text, in the captions of Figures 1,2, and 3.

37) Figure 1: are these figures with or without adaptation?

- Figure 1 does not include adaptation – this has been clarified in the caption.

38) Figure 1: For soy this figure shows yield losses larger than 100%. That seems to make no sense. How are these results handled in the subsequent analysis?

- Yield losses are adjusted to not exceed 99% in subsequent analyses. This has been specified in the figure caption and also in the Methods section.

39) P7L144: It would be good to use a little less econ jargon here. Maybe at least say equivalent variation of income, or something like that.

- This phrase has been changed to “The sum of these three is the total regional welfare change, reported here as real income.” In the case of the Cobb Douglas regional utility function used here, they are equivalent.

40) P9L173: Which version of the FUND model was used?

- Version 3.8 was used for the damage component. This information has been added to the main text.

41) P10L184: What does “damage module based on the FUND model” mean? Are the authors not using FUND for this study?

- See also response to comment 35.
- We are using a standardized socio-economics and climate scenarios. This allows for standardized comparisons of the representation of damages across all three models used for the calculation of the SCC by the US government. Although the initial manuscript only contained values from FUND, the revised manuscript also contains the SCC derived from the DICE damage module and the PAGE market impacts damage module. Using common representations of the socio-economics and climate allows means differences in the SCC can be interpreted as arising from differences in the representation of climate damages.

42) P10L188: The authors should mention here what discount rate is used for these numbers.

- The following sentence was added: “Results presented here use a 3% discount rate. Additional results based on a 2.5% and 5% discount rate are given in Table S4.”

43) P10L190: Could these results include some kind of uncertainty indication? Talking about the SCC in terms of point estimates strikes me as odd.

- Information on the confidence interval for the meta-analysis has been added to the text: “(95% confidence interval based on uncertainty in yield impacts for the meta-analysis is -\$0.6 to \$33.3 per ton)”

44) Figure 3: These figures are really hard to read, especially given that the error bars in some regions use up all the space, so that one can hardly distinguish the central estimates.

- A version of Figure 3 has been added to the supplemental that excludes the error bars, making it easier to distinguish the point estimates of the 3 damage functions. This is referenced in the caption of Figure 3 in the main text.

45) P11L204-5: Don’t use the plural for “damage functions”, the paper only shows things for the FUND damage function.

- The plural was intended to refer to the 16 different agricultural damage functions in each of the FUND regions.
- The statement has been clarified to read “In contrast to existing regional damage functions, which show benefits in every region up to at least 3°C of warming, we find potential for welfare declines even at much lower levels of warming.”

46) P13L249: In general, I assume everything here is based on yearly averages? Would be good to mention this somewhere.

- The referenced statement refers to the calculation of average growing-season temperature, which is an average of 1979-2013 growing-season temperatures, as described in the previous sentence.

47) P17L348: Spelling.

- This has been corrected.

48) P18L370: Why is 3 cited here?

- This reference implements the approach used here of coupling damage modules from the IAMs used for the IAWG report with a common socio-economic scenario and climate model in order to isolate the effect of damages in driving the SCC. This is also the approach used in Rose et al. (2014). Since reference 3 is still a working paper, the reference has been changed to Rose et al. (2014).
- See also response to comment 14.

49) P18L370: If the model is driven by multiple scenarios, how are they aggregated?

- Multiple scenarios are not used. The plural referred to different socio-economic and emissions trajectories in each region. This has been clarified in the text: “The damage module is driven by a standardized socio-economic and emissions pathway(Rose *et al* 2014).”

50) P18L371: What is the standardized climate module used here? I thought the interagency workgroup just picked one distribution for the climate sensitivity, but the text here suggests more standardization?

- We thank the reviewer for pointing this out – this was a mistake in the earlier draft. The IAWG used standardized socio-economic and emissions drivers (as is done in this paper), a distribution over the

climate sensitivity, but the climate models from each model. Our approach uses additional standardization, following Rose et al. (2014), in order to isolate the effects of the damage function in driving differences between models in the SCC. This allows us to say what fraction of market impacts (PAGE) or non-sea level rise impacts (DICE) must come from the agricultural sector, given the new estimates in this paper (see also response to comment 41).

- We have clarified this point and added additional details on the standardized climate model used: “We use a business-as-usual emissions scenario (Scenario 2 in ref. 3), paired with the DICE climate module.” (Methods, p. 23)

51) P19L372: Just above it says that multiple scenarios were used, this sentence suggests that one BAU scenario was used. The reference seems incorrect here, so it is not clear what scenario is used.

- We thank the reviewer for this comment as the reference for the emissions scenario was incorrect. This has been corrected to reference the IAWG report, which documents the emissions scenario used.
- Only one emissions scenario was used, and this has been clarified in the text (see response to comment 49).

52) Figure 4: it is not clear to me how uncertainty is handled in the computation of the SCC. Is the error bar for the meta analysis just based on uncertainty about the agricultural impacts? That doesn't seem to make sense to me, doesn't FUND also have uncertainty about many other parameters?

- Uncertainty bars are only based on uncertainty in agricultural impacts. FUND does include probability distributions over many parameters, but we report results using only the central estimates.
- The meaning of the uncertainty bars, and the distinction with the full Monte Carlo implementation of FUND, have been clarified at two points in the main text: “These agricultural damage functions are then incorporated into a sectorally- and regionally-disaggregated SCC damage module based on the FUND model, keeping the rest of the impact sectors unchanged (Anthoff and Tol 2014, Rose *et al* 2014). Damage functions in the module use the central parameter estimates of FUND. The full FUND model includes probability distributions over many parameters and is designed to be run in a Monte Carlo mode. (Anthoff and Tol 2014) This uncertainty is not dealt with in this paper, meaning uncertainty reported in the SCC reflects only the statistical uncertainty in the yield response from the meta-analysis.” – Methods, p.23 “Uncertainty bars give the SCC resulting from the 95% confidence interval of yield response parameter estimates from the meta-analysis.” – Caption Figure 4, p.13

Reviewer 2

1. Criteria for evidence

For this analysis to be insightful beyond the very specific case presented here, there needs to be a very clear argument about what type and standard of evidence should be used to calibrate the damage functions in a climate economic model. I think that is missing. The authors use a meta-analysis from Chaliner et al. (2014), and all the strengths and weaknesses of that survey of the literature are then absorbed by the current analysis.

This meta-analysis, even in the original paper, doesn't provide us with much critical guidance on what criteria there are for inclusion. For the current paper: What are the criteria for inclusion of the point estimates? How many of them are extensively empirically calibrated or validated? The original article claims some are, but the level of validation appears to vary upon inspecting papers cited therein. How many purely empirical models are there? (Chaliner et al. states that only 4 studies have purely empirical estimates and there does appear to be some systematic differences between these and the process model estimates.) A point estimate from a crop model will use biophysical parameters about a location, but we are left with no sense of whether these models are doing a good job of actually replicating reality, or of their skill in their projections. Most importantly, we have no sense of the consistency across these studies, and it is here where a major contribution could be made – through a systematic effort to standardize the evidence across all of these estimates and to justify it broadly.

Quite a few of these estimates are also over 20 years old, and the criteria for inclusion seems to be just that they have been published. Chaliner et al. note that these estimates may not represent the most up to date science themselves. The number of estimates is also misleading. The top 6 studies cumulatively give about 700 of these point estimates! The consistency and methods across studies is also opaque. Chaliner et al. mention this in terms of adaptation (which I discuss below) but it is likely to be true for a number of different facets of these papers. What are the assumptions, parameters, and calibrations that went into the models used? Was any effort made either by the authors doing the meta-analysis or the current authors to standardize or at least catalog the differences? This could be crucial, not only for adaptation, but also for assumptions about local representativeness (discussed below), credibility of management practices, etc.

An attempt is made to compare the current meta-analysis to the GGCM dataset as validation. And yet this merely compares a set of models to another set of models which themselves have not been extensively validated. To date, I know of no published paper which systematically compares the GGCM to observed data. One that I have been referred to by members of the AgMIP group is currently in submission (Muller et al, available here: <http://www.geosci-model-dev-discuss.net/gmd-2016-207/#discussion>), and finds that the GGCM appears to have a number of limitations when evaluated systematically for predictive skill.

All of this leads me to ask: what is this current evidence representing, and how do we evaluate it for quality? For the current analysis to be the contribution that it claims to be, it needs to be a lot more critical and precise about what counts as evidence. Is a damage function better because it has more data behind it, or should we require that it is closer to the truth? I think the authors need to provide much more detail and transparency on the methodologies behind the point estimates included.

- Substantial additional details on the database of studies used for the meta-analysis has been added to the main text and to the supplementary information:
 “For the four crops, the database contains 1010 observations (344, 258, 336, and 92, for maize, rice, wheat and soybeans respectively) from 56 different studies (many studies report multiple yield changes either for different crops, different locations, different levels of temperature change, or different assumptions about adaptation). The studies include 8 empirical studies and 48 process-based studies, published between 1997 and 2012. Figures S14 and S15 show the geographic coverage of production areas within the database and the distribution of publication dates. Of the 1010 data points, 451 are reported as including some form of on-farm, within-crop, agronomic adaptation. The vast majority of these adaptations involve adjusting either planting date (10%) or cultivar (12%) or both (44%).” – Methods p. 15
- We have included additional robustness checks to investigate whether differences in the studies included significantly affects our estimate of the yield response to temperature change.
 - o To address the reviewer’s concern that process-based models may be inaccurate and not well-founded on empirical data, we allow the effect of warming to differ between process-based and empirical studies. This exercise is somewhat constrained because of the limited number of empirical data-points in the database, particularly at higher temperatures. But we do not find strong evidence that, controlling for other relevant differences between studies, the empirical studies give substantially different results to those based on empirical relationships between yields and climate variability. These results are reported in Figure S17.
 - o We split out the more recent studies (published 2005 or later) and re-estimate our main equation. Figure S16 shows the comparison between the temperature response curves estimated on this subset of studies. We do not find excluding the oldest studies in the database substantially alters our conclusions, particularly given the statistical uncertainties already accounted for in the full analysis. There is some evidence that newer studies show a more negative effect of warming on rice yields, but this is well within our confidence intervals.
- See also response to comment 4 for a more general description of the criteria guiding this approach to updating the damage function

2. Treatment of spatial heterogeneity in the analysis

The treatment of spatial heterogeneity within the analysis is not particularly transparent. The authors state, and one would expect, that differences across space matter. The approach used in the paper abstracts away from this concern. If I follow it correctly, the authors first gather location specific estimates from papers. As above, it is unclear how specifically tailored to differing locations each crop model estimate is. These are then collapsed into a single impact conditional on projected temperature changes, which explicitly disregards spatial heterogeneity. The fact that in observed data there is substantial differences in yields and marginal damages from temperature is then lost in the analysis. These global, non-spatially heterogeneous impact functions are then applied to spatially heterogeneous warming to give crop impacts that are

heterogeneous across space. This heterogeneity in TEMPERATURE is then used to model the equilibrium adjustments under trade. Finally, these welfare impacts are then aggregated up to the regions of FUND, where FUND's own model will apply regionally heterogeneous growth and climate projections (though the former appear to be based on a linear scaling of a global growth projection that preserves the (presumably) year 2000 distribution of global income).

This would not be especially troubling were it not for the fact that the regional results are presented in the paper as meaningful, but tracing the exact source of those regional differences seems difficult and untransparent. For the spatial aspect of this to be included in the paper, I would expect a much more detailed treatment (along with sensitivity analysis) of where each transformation of the initial global impact coefficient occurs, and how much distortion arises from choosing a globally representative coefficient.

- Spatial heterogeneity in **yield** impacts doesn't come only from spatial heterogeneity in warming but also from differences in baseline temperature through the interaction between baseline temperature and warming (parameters β_{3j} and β_{4j} in Equation 1 in the revised manuscript). The same amount of local warming is allowed to have different effects in cold compared to hot places.
- The meta-analysis approach does deliberately average over what might be real spatial differences in the marginal response to temperature or CO2 in the interests of more precisely estimating an average value (for example due to soil variability or management practices). This is part of the reason why we present the GGCMI as a comparison – this set of models captures a much richer set of spatially-varying variables (e.g. soil, management (in some cases), rainfall) and therefore can be used as a comparison.
- Our ability to distinguish additional spatial heterogeneity in the yield response to warming is constrained by data availability. We are already differentiating the marginal response to warming by crop, by level of warming (i.e. the quadratic term), and by average growing-season temperature. In total this uses 16 degrees of freedom. Although there are 1010 observations, these come from 56 studies. Since we allow for correlation between observations from the same study, this limits our ability to add additional interaction terms to the temperature response.
- Spatial heterogeneity in the **economic** impacts of yield changes (calculated using GTAP and then used as input to FUND) arises from multiple sources:
 - Differences in yield impacts arising from variation in local warming and variation in baseline growing season temperature as discussed above.
 - Differences in the baseline mix of crops in between countries and differences in their extensive and intensive margins of supply response
 - Differences in a country's trade position with respect to different crops
 - Differences in consumption preferences between countries, including the current pattern of consumption as well as their price elasticities of demand

- Differences in agricultural market distortions (subsidies and taxes) between countries, as the climate-induced yield shocks interact with these distortions to result in second-best welfare effects (Hertel and Randhir 2000)
- The first heterogeneity is captured in the yield modeling step (or AgMIP results) and the others in the GTAP model.
- Explanations have been added within the main text clarifying sources of spatial heterogeneity for the yield changes:
 “The two approaches for estimating yield impacts differ substantially in the kinds of spatial heterogeneity in the yield response to warming and CO₂ that are captured. The GGCM results explicitly account for spatial variation in the yield response to temperature change resulting from soil-type, irrigation, baseline temperature, and (in models representing nitrogen stress) nutrient limitations. The meta-analysis deliberately smooths out most of this heterogeneity in order to more precisely estimate a common response function, preserving only the heterogeneity resulting from different baseline temperature.” – Results, p.7
- A table has been added to the supplemental information (Table S2) describing the spatial heterogeneity captured at each stage of the analysis.

3. Adaptation as a blackbox

The way the paper is written, it appears to be dealing with adaptation. However, it deals with only some forms, often inconsistently, and doesn't track the impact of different adaptation assumptions through the analysis. What determines whether adaptations occur? Are all farmers rational optimizers and so automatically adjust their planting behavior immediately? Do we observe any of the same adaptations in real data? Are these adaptations comprehensive? It seems to be limited to just marginal changes in cropping behavior and not adaptations based on changing crops, exit from agriculture, general shifts in cropping patterns, etc. In fact, it seems there is much left out, and that calling this adaptation overstates what is being included in the analysis. This needs to be made much clearer in the early stages of the paper, since the impression is of more comprehensiveness than actually is included.

Does each of the studies deal with adaptation in the same way – is there a unified treatment of adaptation or an adaptation scenario within the estimates? Reading through the discussion of adaptation in the meta-analysis of Chalinor et al, it appears not. They point out that, “only two categories [of adaptation behaviours] have more than 20 entries in the paired adaptation studies”. This part of the exercise becomes problematic.

There's also a few layers of adaptation here: minute marginal adjustments treated inconsistently across agronomic models (and presumably included implicitly in the quantitatively distinct statistical estimates in Chalinor et al), adjustments of production through a GE calculation in GTAP, and then the adaptation that is embedded into FUND itself. As far as I can tell, none of these sources of adaptation is well-founded in actual

data, so using the term in the paper results in a complex blackbox. The authors should be very clear about the types of adaptation being discussed throughout, whether any of these have been observed in actual data as occurring, and treat them all coherently and consistently (i.e., by looking at alternative cases using ONLY adaptations of a particularly type, not a single dummy in the main analysis that groups ALL of the limited set of adaptations together). Importantly, they should give a detailed description of the types of adaptation that are not being considered, as currently the broad term “adaptation” is used without clarification that this leaves out some of the most important proposed adaptive behaviors.

If it is possible to reanalyze the papers in the meta-analysis to be consistent across studies, then something more can be said in the paper. Failing being able to treat this in a consistent way across all estimates and all steps of the process (including alternate analyses tracking the contributions of each to the final SCC), I feel it would be better to be clear about the limitations in the analysis, state that this will deal only with exactly quantifiable and consistent forms of adaptation or with none at all, and have an un-adaptation adjusted estimate of damages and the SCC as the main result. In my opinion, that is a more transparent approach than what is currently employed.

- We thank the reviewer for this comment and agree that the description of adaptation was underdeveloped in the original manuscript. In particular, as the reviewer points out, there are several types of adaptation that are relevant: on-farm, within-crop agronomic adaptations (e.g. adjusting planting date and cultivar); on-farm economic responses to price and productivity changes (e.g. crop switching, intensification of production); and other adjustments in the agricultural sector that reduce the impact of productivity shocks (e.g. adjusting trade patterns, consumption switching). Only the first type is dealt with in the meta-analysis. The others are dealt with through the GTAP modeling. Previously the term “adaptation” was used in the text to refer only to the within-crop agronomic adaptations, and this has been clarified.
- Throughout the main text, the description of adaptation has been clarified to specify that principally what had been referred to just as “adaptation” is actually on-farm, within crop agronomic adaptations:
“These new damage functions reveal far more adverse agricultural impacts than currently represented in IAMs, as well as a limited potential for agronomic adaptations.” – Abstract, p. 1
“The effect of agronomic, **on-farm, within-crop** adaptations (principally changes in crop variety and planting date (Methods)) is small and statistically insignificant.”– Results, p. 6, changes shown in bold
“Note that this statement refers *only* to the on-farm, within-crop agronomic adaptations captured by the studies that support the meta-analysis. Additional economic adaptations such as crop switching, increasing intensity, substituting consumption or adjusting trade relationships are captured in the GTAP model (Table S5).” – Footnote, results, p. 6
“Of the 1010 data points, 451 are reported as including some form of on-farm, within-crop, agronomic adaptation. The vast majority of these adaptations involve adjusting either planting date (10%) or cultivar (12%) or both (44%).” – Methods p. 15

- Although a decomposition of each type of adaptation to total welfare change would be interesting, it is a major undertaking and we believe it to be beyond the scope of this analysis. It is a direction of research that we intend to pursue further in the future.
- We are able to analyze the effects of changing planting date and cultivars separately, since these make up the bulk of the agronomic adaptation adjustments included in the Challinor et al. database. Rather than including a single adaptation intercept and temperature interaction, we include two – one for studies that include changing planting date, and one for studies that include changing cultivar. The results are documented in Table S6. Thought point estimates do differ between adaptation types, standard errors are large and we cannot reject the hypothesis either that they are equal to zero or equal to the common effect estimated in our preferred model.
- We have included a table in the SI (Table S5) that describes the different types of adaptation included at each stage of modeling.
- We do not agree that an estimate not including adaptation would be preferred for several reasons:
 - 1) Theoretically, damage functions used to estimate the SCC should include the net benefits of adaptation (for instance Cropper and Oates 1992, Stern 2006, p 405, Mendelsohn *et al* 1994). This is particularly true, as is the case for most agronomic adaptations, when adaptation confers a private good.
 - 2) Not including on-farm agronomic adaptations while still including other economic adaptations as part of the GTAP modeling would be inconsistent and solving the economic model without adaptive economic behavior would be computationally problematic.
 - 3) Although identifying adaptation is challenging, on-farm adaptations consistent with climate change already experienced have been documented. For instance Menzel et al. (2006) show planting dates in Germany have been advancing (although at a slower pace than implied by natural phenological changes), Seifert and Lobell (2015) show an expansion of double-cropping in the areas of the US where climate warming has expanded the growing season, and Kucharik (2006) documents a multi-decadal advancement of corn planting dates in the US.

4. Does the paper establish a broad and generalizable approach to updating climate damages?

The overall question that I am left with is: does the currently proposed methodology provide sufficient guidance for a much-needed overhaul of the way that we in the IAM community deal with damages? While interesting, I think it is hard to draw the conclusions needed to do so. The headline number on the SCC may provide an interesting outcome of the analysis, but it is of less importance than the potential

methodological contribution of providing a set of standards and criteria for what evidence to incorporate into the SCC calculation. It is generally accepted in the community at the moment that guidance on what evidence to include is of more importance than adjusting damage functions with no general guidance. It is my opinion that re-orienting the paper to be this kind of contribution is a worthwhile if substantial exercise.

- The guiding principle is that the most legitimate, efficient, and generalizable approach is to follow, to the degree possible, the findings of the IPCC. This has the benefits that findings are updated on a moderately regular basis (every 7 years), are assembled and reviewed by impact experts within each field, and are formally accepted as fact by governments (including the US). To the extent this process produces estimates that can be used to inform IAM damage functions, we believe they should form the empirical basis of new SCC estimates.

- In addition, both the meta-analysis and AgMIP damage functions reflect an ‘ensemble’ principle – it has been shown both in climate modeling and, increasingly, in agricultural modeling, that use of multi-model ensemble averages tends to out-perform any individual model (IPCC 2013, p 766, Asseng *et al* 2014). Therefore, averaging over multiple model outputs should lead to both more reliable damage functions and better uncertainty quantification than trying to pick preferred models.

- Finally, we note that in mid-January (after the manuscript was submitted and reviewer comments written) a review by the National Academy of Sciences on the methods used by the US government to estimate the SCC has provided principles that should guide improving the SCC. These principles are that improvements should be: 1) consistent with the scientific basis as represented in the current peer-reviewed literature; 2) uncertainties should be characterized and quantified where possible; and 3) the process should be transparent, well-documented and reproducible. We believe this paper to be consistent with all three of these criteria (see also response to Reviewer 1, comment 10 regarding transparency and reproducibility) (NAS 2017, pp 8–9).

- A paragraph in the introduction has been substantially revised to incorporate this information: “This approach exemplifies several principles that we believe can provide important guidance in updating damage functions in other sectors in the future. Our meta-analysis is related to the findings of the food security chapter of the IPCC 5th Assessment Report. Where possible, tying damage functions to the IPCC has several benefits: findings are updated on a moderately-regular basis (every 7 years), are assembled and reviewed by impact experts within each field, and are formally accepted as fact by governments involved in the process. Both the meta-analysis and the AgMIP damage functions also use an ensemble of models. Findings in climate and, increasingly, agricultural modeling have shown that use of multi-model ensembles tends to out-perform any individual model (IPCC 2013, Asseng *et al* 2014). Therefore, averaging of multiple mode outputs should lead to more reliable damage functions and better uncertainty quantification than trying to pick preferred models. Finally, this an “end-to-end”

analysis that directly connects the biophysical science of climate change impacts to the SCC in a single study. This means our damage functions and the changes we identify in the SCC have a clear and traceable connection to the underlying science that is both comprehensive and up-to-date, in contrast to most current IAM damage functions.” – Introduction, p. 5

5. Miscellaneous:

a. The exact critique of the current IAM damages (that they are out of date) is perhaps not the correct one. While in FUND the literature is out of date, as in the other models, the problem is less with the recency of evidence used and more about the type of evidence used, and the standards for inclusion. This perhaps represents more of a cultural drift (between impacts, integrated modelling, and damage communities) than it does any intentional failings in the models. The statements about “20 year old evidence” are less relevant than the fact that the evidence is opaque and somewhat arbitrary.

- Much of this language has been modified in response to comments from Reviewer 1
- In general, we have tried to avoid statements about IAM damage functions in general, given that a detailed review of the supporting literature of damage functions is beyond the scope of this analysis.
- See also response to comments 1 and 4 above regarding standards of inclusion for this analysis.

b. I’m confused as to why the local-to-global warming relationship was used to downscale the mean global temperature. Is there any reason to not use the gridded, spatially heterogeneous output of the CMIP5 ensemble itself? If there are clear reasons for this, it should be made clear in the text. To understand the difference outcome here, it would be helpful to see the current method compared to the gridded model output.

- The CMIP5 ensemble gives how local temperatures change in response to a given emissions scenario. But IAMs have their own emissions scenarios and climate models. The damage function relates a given change in global mean temperature to a change in economic welfare and therefore the CMIP5 output cannot be used directly.
- We do use the spatially heterogeneous output of the ensemble in order to derive the local-to-global warming ratio (Figure S18)

c. For the derivation of the damage function confidence intervals, it is noted that the SEs are calculated to block bootstrapping at the study level. Does this account for all agronomic model interdependencies? Is there any model that produces estimates across studies? If so, that would seem to be a better level at which to resample. This

concern is perhaps a result of the lack of transparency about the methodologies underlying the point estimates in the meta-analysis.

- The meta-analysis does assume independence between different studies, which may be invalid if authors are the same, or if the same crop model is used by different authors. We have added more detail on this assumption and the set of models used to the Methods section: “This treatment of the errors does assume independence between studies, which may be questionable if the same model is used in multiple studies. In total, 28 models are represented in the database, made up of 17 process-based model families² and 11 statistical models, are used in the 56 studies.”

d. The number of included point estimates varies in the paper. Chalinor et al has 1700, and the authors note that this decreases when restricting to the 4 crops used herein, to 1010, but later states that the number is 1030. I suspect it is the latter, since the crop specific numbers on page 13 of the manuscript are consistent with that.

- We thank the reviewer for pointing out this discrepancy. The correct number is 1010 (344, 238, 336, and 92, for maize, rice, wheat and soybeans respectively). The number of observations for rice had been mistakenly reported as 258 when it should have been 238. This has been corrected in the revised manuscript.

e. The authors note that they show systematic differences from the GGCM across the tropics and temperature regions, and then judge this to not be important when aggregated globally. While this is true, the fact that the current paper values regional impacts in welfare terms means that the difference here is of importance. This once again argues for more clarity on how they are tracking spatial heterogeneity throughout the analysis, and whether this reflects a real and meaningful heterogeneity of evidence-based impacts.

- It is true that spatial differences in productivity do matter for regional welfare results. We note the comparison of global averages more as a point of comparison. The regional differences in productivity are carried forward into the welfare results and ultimately into the FUND. The spatial differences between the meta-analysis and AgMIP yield results may help to explain differences in the regional welfare functions. For instance, temperate regions (where the meta-analysis tends to be pessimistic compared to AgMIP) tend to have more positive AgMIP damage function than meta-analysis (e.g. Central and Eastern Europe, Canada, Former Soviet Union).

Reviewer #3 (Remarks to the Author):

I found the topic of the paper interesting. The paper is well written and addresses a relevant question. The authors analyze the sensitivity of the forecasted social cost of carbon to the damage function used to predict climate change effect on

² i.e. treating CERES-maize, CERES-rice and CERES-wheat as a single model

crop yields.

I think that several issues need to be addressed by the authors before publication;

- The authors state that the current damage functions used in integrated assessment models are out of date (for example on line 62), but this is barely documented by the authors. What is the origin of these damage functions? Are these functions the same in all IAM? Is it proven that these "old" functions are inaccurate compared to more recent damage functions ?
- The most relevant damage function is the FUND agriculture sector damage function. The impacts of changing temperature on agriculture in FUND are given by quadratic damage functions for each region. The reference for this is a meta-analysis of five economic studies in Tol (2002b): Kane (1992), Reilley et al (1994), Tsigas (1996), Fischer et al. (1996) and Darwin et al. (1995). These in turn are based on parameterizations of the underlying biophysical impacts. For instance Kane (1992) references Ritchie et al. (1990), Rosenzweig (1990), Peart et al. (1990), and Santer (1985) as their basis for the biophysical impacts of climate change on yields.
- A full accounting of the scientific studies supporting all IAM damage functions is beyond the scope of this analysis. Instead, in the revised manuscript we cite a recent National Academy of Sciences report on the SCC (NAS 2017). In Table 5-2 (p. 188) this reference describes the documented empirical basis of IAM damage functions. The same table is included in EPRI (2014), which is also cited in the manuscript.
- The three IAMs used to calculate the SCC do not have the same damage function. All three models represent different combinations of regions and sectors and so include different damage functions. FUND is the only model to explicitly represent the agricultural sector. This is now included as part of the main text: "Of the three IAMs used to estimate the SCC, only FUND explicitly represents the agricultural sector, so that has been the focus of comparison in this paper. However, agricultural impacts are a part of damages in the two other models. In PAGE (2009 version) agricultural impacts are represented within the market-impacts damage function and in DICE (2013 version) they are in the non-sea level rise damage function (Hope 2011, Nordhaus and Sztorc 2013)." – Discussion p. 14
- The reviewer's final question is what we are trying to answer with this study. To our knowledge there has been no previous comparison of a new agricultural sector damage function with the existing representation in IAMs.
- The authors wrote that they "present a new meta-analysis" (75). But what is really new in their meta-analysis? The dataset was already published by Challinor et al. (2014).
- The results we present here are a new analysis of the dataset published by Challinor et al. (2014). This has been clarified in the revised manuscript: "We start with the large agronomic literature on how climate change affects crop yields based on a meta-analysis published by Challinor et al. (2014) and used to support conclusions in the IPCC 5th Assessment Report (Porter *et al* 2014). We present a new analysis of this database that we use to aggregate these results to the global scale." – Introduction, p.4
- The authors mention that their damage functions are "up-to-date" (85). I am not fully convinced, especially for wheat. For this crop, the authors used 336 yield

data. This is very small compared to the size of the dataset used by Wilcox and Makowski (2014) in their meta-analysis (Wilcox and Makowski. 2014. A meta-analysis of the predicted effects of climate change on wheat yields using simulation studies. *Field Crops Research* 156, 180–190). I advise the authors using this dataset for wheat as it includes more than 1000 wheat yield data.

-
- We thank the reviewer for pointing out this article. The Wilcox and Makowski (2014) paper does provide additional data-points on wheat yield changes in response to changing climate conditions – they report results from 28 articles with 1084 data-points for all three variables. Some of these studies are already reported in the Challinor et al. (2014) database, but many are not.
- While using this database would increase the data available for estimation for wheat yields, we do not believe incorporating the information into this analysis is desirable at this stage. Firstly, there is a strong rationale for tying IAM damage functions to IPCC findings (see response to Reviewer 2, comment 4). Secondly, there are also benefits in terms of both consistency for the economic modeling and uncertainty quantification (in terms of capturing covariance between response parameters for different crops) of estimating all crops in a common framework. Therefore we have not incorporated the Wilcox and Makowski data into our analysis.
- We do note that the findings of responses in both studies are similar: both studies find that CO₂ benefits outweigh the negative effects of warming at lower levels of warming, leading to moderate yield increases, but that temperature effects dominate between 2-3 degrees of warming resulting in yield declines.

- The authors did not explain why they produced a new yield response function instead of using the function already published by Challinor et al. (2014). For wheat, they could also directly use the meta-regression model published by Wilcox and Makowski (2014). Why producing a new meta-regression?

- We reanalyzed the Challinor et al (2014) database for several reasons related to the goal of the study (i.e. producing regional welfare changes for incorporation into IAMs):
 - 1) For the economic modeling it is important to represent as many crops as possible and so we included soy, which is not reported as a separate effect in the original Challinor et al paper
 - 2) We needed the non-linear crop-specific marginal response to temperature change, controlling for changes in other climate variables, which is not reported in Challinor et al. Figure 1 in that paper shows a fit through yield changes as a function of temperature change, but does not control for other climate variables. If there is something systematically different about studies at different parts of the temperature distribution, this fit can not be interpreted as the partial effect of a change in temperature. (For instance empirical studies disproportionately report estimates at 1°C of warming, but also disproportionately do not include the CO₂ effect, biasing the unconditional relationship between yield and temperature changes in the database). Table 1 in Challinor et al. (2014) does give the results of a multiple regression, but the specification is too coarse for our purpose – crop-specific temperature effects are not reported and all relationships are linear in climate variables.
 - 3) In response to comments on earlier versions of the paper (see also Reviewer 2, comment 2), we increased the spatial heterogeneity of the yield response from the initial temperate / tropical distinction in Challinor et al (2014) by introducing the interaction with baseline temperature. This is important for the input to the disaggregated GTAP model.

- Using the model reported in Wilcox and Makowski (2014) would not be desirable for this application because they estimate a random-effects model. This allows the response to changes in climate variables to differ between locations but does not explicitly model this heterogeneity, meaning it is impossible to extrapolate beyond the set of locations included in the study. Input to GTAP requires extrapolation of productivity changes to a global grid which is not possible with the random-effects model.

- In lines 108-112, the authors explain that the CO₂ function does not produce realistic values. I advise the authors using a different function. For example, they could replace their $f(\text{CO}_2)$ function by a quadratic function whose parameters could be directly fitted to the data. They could also try the meta-regression models mentioned above.

- The methods have been modified in response to Reviewer 1 to include a CO₂ fertilization effect for maize in the meta-analysis. The effect is allowed to differ from that of C₃ crops, but is of similar magnitude.

- Before selecting the chosen functional form, we considered two alternatives, both found to be inferior to the chosen form. A quadratic form is inconsistent with theory because it shows a negative marginal effect of CO₂ at high temperatures. This is because most of the data falls around 560ppm when warming at 3-5 degrees results much higher concentrations, which fall past the optimum of the estimated effect.
- We also investigated a logarithmic response function. This is not preferred because there was found to be a very strong dependence on the point chosen as baseline CO₂, related to the behavior of the logarithmic function close to zero.
- The chosen functional form allows flexibility over both the saturation point (fit through the regression) and the rate at which the saturation point is reached (through the “A” parameter”).
- The original Chaliner et al. (2014) meta-analysis used a linear response to changing CO₂, but we believe this to be inappropriate for extrapolating out to large CO₂ changes associated with 2-3 degrees of warming as is required for this analysis.

- The statistical model is presented on page 13, but the authors do not explain how this model was selected. This model should be compared to other models using standard statistical methods. In addition, the authors should provide the readers with all parameter values. Only three parameter values are presented in Table S1. Why? Finally, the authors should analyze the distribution of model residuals in order to check whether the assumptions made by their model are realistic.

- The chosen model is not attempting to provide the best out-of-sample prediction or explanation of sample variance (both standard model-selection criteria). Rather it is derived from theory, with a focus on the ultimate goal of the exercise: informing estimates of how temperature changes affect regional welfare changes through their effect on the agricultural sector. This is why a large fraction (16 out of 21) parameters estimated are used to describe heterogenous response to warming, which is allowed to differ by crop, by level of warming (quadratic terms) and by baseline growing season temperature. These terms may not be selected

using model selection, but they are nevertheless theoretically important for informing the GTAP analysis (e.g. see Reviewer 2, comment 2).

- The three (now 4 in the revised manuscript) parameters reported in the parameter table are all the non-temperature parameters (other than the adaptation intercept term which is not used in the analysis, but is now reported in Table S6 in response to Reviewer 2 comment 3).

- The temperature parameters are not reported because 1) interpretation of coefficients of non-linear functions (particularly those including interaction terms as is the case here) is not intuitive; and 2) standard errors of those coefficients are not meaningful because the uncertainty of the full response function also depends on correlation between coefficients. This is why we instead report the 16 temperature coefficients (β_{1j} , β_{2j} , β_{3j} , β_{4j}) as well as the uncertainty associated with them graphically in Figure 1.

- We use a non-parametric block-bootstrap to estimate the uncertainty bounds that we propagate through the GTAP model and subsequently the SCC. Therefore our inference does not rely on any distributional assumptions of residuals.

- In eq.(1), the effects of some variables are crop-dependent, while the effects of others are the same for all crops. Why?

- Given limited data (particularly given we allow for correlation between observations from the same study), the statistical power to precisely identify crop-specific effects is limited. Since the purpose of the paper is to produce a temperature damage function, we use that statistical power to produce temperature response functions that vary by crop and baseline temperature. But the limited dataset means we are not able to do the same for the other variables.

References:

- Anthoff D and Tol R S J 2014 FUND v3.8 Scientific Documentation
- Appeals U C of 2016 *Zero Zone, Inc. et al., v. United States Department of Energy, et al.* Online: <http://media.ca7.uscourts.gov/cgi-bin/rssExec.pl?Submit=Display&Path=Y2016/D08-08/C:14-2159:J:Ripple:aut:T:fnOp:N:1807496:S:0>
- Asseng S, Ewert F, Martre P, Rötter R P, Lobell D B, Cammarano D, Kimball B A, Ottman M J, Wall G W, White J W, Reynolds M P, Alderman P D, Prasad P V V., Aggarwal P K, Anothai J, Basso B, Biernath C, Challinor A J, De Sanctis G, Doltra J, Fereres E, Garcia-Vila M, Gayler S, Hoogenboom G, Hunt L A, Izaurrealde R C, Jabloun M, Jones C D, Kersebaum K C, Koehler A-K, Müller C, Naresh Kumar S, Nendel C, O'Leary G, Olesen J E, Palosuo T, Priesack E, Eyshi Rezaei E, Ruane A C, Semenov M A, Shcherbak I, Stöckle C, Stratonovitch P, Streck T, Supit I, Tao F, Thorburn P J, Waha K, Wang E, Wallach D, Wolf J, Zhao Z and Zhu Y 2014 Rising temperatures reduce global wheat production *Nat. Clim. Chang.* **5** 143–7 Online: <http://dx.doi.org/10.1038/nclimate2470>
- Burke M, Craxton M, Kolstad C D, Onda C, Allcott H, Baker E, Barrage L, Carson R, Gillingham K, Graff-Zivin J, Greenstone M, Hallegatte S, Hanemann W M, Heal G, Hsiang S, Jones B, Kelly D L, Kopp R,

- Kotchen M, Mendelsohn R, Meng K, Metcalf G, Moreno-Cruz J, Pindyck R, Rose S, Rudik I, Stock J and Tol R S J 2016 Opportunities for advances in climate change economics *Science (80-.)*. **352** 292–3 Online: <http://science.sciencemag.org/content/352/6283/292.abstract>
- California S of 2016 *Assembly Bill 197* Online:
https://leginfo.legislature.ca.gov/faces/billNavClient.xhtml?bill_id=201520160AB197
- Carleton T A and Hsiang S M 2016 Social and economic impacts of climate *Science (80-.)*. **353** 1112
- Challinor A J, Watson J, Lobell D B, Howden S M, Smith D R and Chhetri N 2014 A meta-analysis of crop yield under climate change and adaptation *Nat. Clim. Chang.* **4** 287–91
- Cropper M L and Oates W E 1992 Environmental Economics: A Survey *J. Econ. Lit.* **30** 675–740
- Darwin R, Tsiga M, Lewandowski J and Ranases A 1995 *World Agriculture and Climate Change: Economic Adaptations* Online: <http://ideas.repec.org/p/ags/uerser/33933.html>
- DeVuyst E A and Preckel P V. 1997 Sensitivity Analysis Revisited: A Quadrature-Based Approach *J. Policy Model.* **19** 175–85
- Fischer G, Frohberg K, Parry M L and Rosenzweig C 1996 Impacts of Potential Climate Change on Global and Regional Food Production and Vulnerability *Climate Change and World Food Security* ed T Downing (Berlin: Springer-Verlag) pp 115–59
- Hertel T W and Randhir T O 2000 Trade Liberalization as a Vehicle for Adapting to Global Warming *Agric. Resour. Econ. Rev.* **29** 159–72
- Hope C W 2011 *The PAGE09 Integrated Assessment Model: A Technical Description*
- Huff K and Hertel T W 1996 *Decomposing Welfare Changes in GTAP* (Purdue, IN)
- IAWG U 2013 Technical support document: Technical update of the social cost of carbon for regulatory impact analysis under executive order 12866 1–22
- IPCC 2013 *Climate Change 2013: The Physical Science Basis. Contribution of Working Group 1 to the Fifth Assessment Report of the Intergovernmental Panel on Climate Change* ed T F Stocker, D Qin, G K Plattner, M Tignor, S K Allen and A Boschung (Cambridge, UK: Cambridge University Press)
- IPCC 2014 Summary for Policymakers *Climate Change 2014: Impacts, Adaptation and Vulnerability. Working Group 2 Contribution to the IPCC 5th Assessment Report* ed C B Field, V R Barros, D J Dokken, K J Mach, M D Mastrandrea, T E Bilir, M Chatterjee, K L Ebi, Y O Estrada, R C Genova, B Girma, E S Kissel, A N Levy, S MacCracken, P R Mastrandrea and L L White (Cambridge, UK: Cambridge University Press)
- Kane S, Reilly J and Tobey J 1992 An Empirical Study of the Economic Effects of Climate Change on World Agriculture *Clim. Change* **21** 17–35
- Knutti R and Sedláček J 2012 Robustness and uncertainties in the new CMIP5 climate model projections *Nat. Clim. Chang.* **3** 369–73 Online: <http://www.nature.com/doi/10.1038/nclimate1716>
- Kucharik C J 2006 A multidecadal trend of earlier corn planting in the central USA *Agron. J.* **98** 1544–50
- Larson A 2016 Subsidies Proposed for New York's Upstate Power Plants *Power* Online:
<http://www.powermag.com/subsidies-proposed-for-new-yorks-upstate-nuclear-power-plants/>
- Leakey A D B, Ainsworth E A, Bernacchi C J, Rogers A, Long S P and Ort D R 2009 Elevated CO₂ effects on plant carbon, nitrogen, and water relations: six important lessons from FACE *J. Exp. Bot.* **60** 2859–76
- Mendelsohn R, Nordhaus W D and Shaw D 1994 The Impact of Global Warming on Agriculture: A Ricardian Analysis *Am. Econ. Rev.* **84** 753–71
- Menzel A, Von Vopelius J, Estrella N, Schleip C and Dose V 2006 Farmers' Annual Activities are not Tracking the Speed of Climate Change *Clim. Res.* **32** 201–7
- Morita T, Kainuma M L T, Harasawa H, Kai K, Dong-Kun L and Matsuoka Y 1994 *Asian-Pacific Integrated Model for Evaluating Policy Options to Reduce Greenhouse Gas Emissions and Global Warming Impacts* (Tsukuba)
- MPUC 2016 *In the Matter of the Further Investigation into Environmental and Socioeconomic Costs*

- Under Minnesota Statutes Section 216B.24222, Subdivision 3* Online:
https://mn.gov/oah/assets/2500-31888-environmental-socioeconomic-costs-carbon-report_tcm19-222628.pdf
- Mueller N D, Gerber J S, Johnston M, Ray D K, Ramankutty N and Foley J A 2012 Closing yield gaps through nutrient and water management *Nature* **490** 254–7 Online:
<http://www.nature.com/doi/10.1038/nature11420>
- NAS 2017 *Valuing Climate Damages: Updating Estimation of the Social Cost of Carbon Dioxide* (Washington, D.C.)
- Nordhaus W D Revisiting the social cost of carbon
- Nordhaus W D and Sztorc P 2013 *DICE 2013R: Introduction and User's Manual* Online:
<http://scholar.google.com/scholar?hl=en&btnG=Search&q=intitle:DICE+2013R++Introduction+and+User's+Manual#0>
- Pearl R, Jones J and Curry R 1990 Impact of Climate Change on Crop Yield in the Southeastern U.S.A. *The Potential Effects of Climate Change on the United States: Report to Congress* (Washington, D.C.)
- Pindyck R S 2013 Climate Change Policy: What Do the Models Tell Us? *J. Econ. Lit.* **51** 860–72 Online:
<http://pubs.aeaweb.org/doi/abs/10.1257/jel.51.3.860>
- Porter J R, Xie L, Challinor A J, Cochrane K, Howden M, Iqbal M M, Lobell D B and Travasso M 2014 Chapter 7: Food Security and Food Production Systems *Climate Change 2014: Impacts, Adaptation and Vulnerability. Working Group 2 Contribution to the IPCC 5th Assessment Report* (Cambridge, UK: Cambridge University Press)
- Reich P B, Hobbie S E and Lee T D 2014 Plant growth enhancement by elevated CO₂ eliminated by joint water and nitrogen limitation *Nat. Geosci.* **7** 920–4 Online:
<http://www.nature.com/doi/10.1038/ngeo2284>
- Reilly J, Hohnmann N and Kane S 1994 Climate Change and Agricultural Trade: Who Benefits and Who Loses? *Glob. Environ. Chang.* **4** 24–36
- Revesz R, Arrow K, Goulder L, Kopp R E, Livermore M, Oppenheimer M and Sterner T 2014 Improve Economic Models of Climate Change *Nature* **508** 173–5
- Ritchie J, Gaer B and Chou T 1990 Effect of Global Climate Change on Agriculture: Great Lakes Region *The Potential Effects of Climate Change on the United States: Report to Congress1* (Washington, D.C.)
- Rose S, Turner D, Blanford G J, Bistline J, de la Chesnaye F and Wilson T 2014 *Understanding the Social Cost of Carbon: A Technical Assessment* (Palo Alto, CA) Online: <http://www.epri.com/>
- Rosenzweig C 1990 Potential Effects of Climate Change on Agricultural Production in the Great Plains: A Simulation Study *The Potential Effects of Climate Change on the United States: Report to Congress1* (Washington, D.C.)
- Rosenzweig C, Elliott J, Deryng D, Ruane A C, Müller C, Arneth A, Boote K J, Folberth C, Glotter M, Khabarov N, Neumann K, Piontek F, Pugh T A M, Schmid E, Stehfest E, Yang H and Jones J W 2014 Assessing agricultural risks of climate change in the 21st century in a global gridded crop model intercomparison. *Proc. Natl. Acad. Sci. U. S. A.* **111** 3268–73 Online:
<http://www.pnas.org/content/111/9/3268>
- Santer B 1985 The Use of General Circulation Models in Climate Impact Analyses - A Preliminary Study of the Impacts of a CO₂ Induced Climate Change on West European Agriculture *Clim. Chang.* **7**
- Schlatter L 2016 *Findings of fact, conclusions, and recommendations: carbon dioxide values* (Saint Paul, MN)
- Seifert C A and Lobell D B 2015 Response of double cropping suitability to climate change in the United States *Environ. Res. Lett.* **10** 24002 Online: <http://stacks.iop.org/1748-9326/10/i=2/a=024002?key=crossref.615c30b00a05e112de05663c69ac28b7>
- Stern N 2016 Economics: Current climate models are grossly misleading *Nature* **530** 407–9 Online:

- <http://www.nature.com/news/economics-current-climate-models-are-grossly-misleading-1.19416>
- Stern N 2006 *The Economics of Climate Change: The Stern Review* (Cambridge: Cambridge University Press)
- Tol R S J 2002a Estimates of the Damage Costs of Climate Change, Part II. Dynamic Estimates *Environ. Resour. Econ.* **21** 135–60 Online: <http://link.springer.com/10.1023/A:1014539414591>
- Tol R S J 2002b Estimates of the damage costs of climate change. Part I: Benchmark estimates *Environ. Resour. Econ.* **21** 47–73 Online: <http://link.springer.com/article/10.1023/A:1014500930521>
- Tsigas M E, Frisvold G B and Kuhn B 1996 Global Climate Change in Agriculture *Global Trade Analysis: Modelling and Applications* ed T Hertel (Cambridge: Cambridge University Press)
- Wilcox J and Makowski D 2014 A meta-analysis of the predicted effects of climate change on wheat yields using simulation studies *F. Crop. Res.* **156** 180–90 Online: <http://dx.doi.org/10.1016/j.fcr.2013.11.008>

Reviewers' comments:

Reviewer #2 (Remarks to the Author):

First, I want to commend the authors on all of the effort put into the revision of this paper. It does reclaim a lot of the inner workings from a type of "black-box" territory, which was the general theme of my previous comments. In particular, I found the supplementary materials on adaptation to be helpful. I still, however, find that the overall approach may not be good guidance for how to use new evidence to update, though the set of methods for this particular problem and data are more open in this revision.

One general comment is to make an effort to remove any remaining unsubstantiated negativity about the current fleet of IAMs (as referee 1 pointed to frequently). I present additional comments to the current draft below in no particular order.

1. The comment about 20 year old science should be removed from the abstract. The first papers in the current metaanalysis are also 20 years old. Again, it is not about recency, but the fact that they have been superseded by other results and they are not transparent.

2. If this is being pitched as a general issue / approach, with an underlying scientific support from the IPCC, then some mention should be made of the evidence in the IPCC on other sectors.

3. In the regression, noting that there are 28 models, the standard errors should probably be clustered on model rather than on study. At least as a robustness check.

4. The paper has no citation to Houser et al (2014) (Economic Risks of Climate Change in America), which appears to have taken an empirical approach to damage function estimation not dissimilar to the current paper, though for multiple sectors. This should be included.

5. My previous comment 3 asked whether any of the adaptations had been observed in historical data. As a clarification, I was asking this specifically for the adaptations in the process models. Expanding on that, I wish to know how close to reality the adaptation parameters are within those models, not just if a similar adaptation has been observed. In my experience, many process models overstate the benefits from these choice parameters.

6. On the quality of the evidence, more detail should be given on the criteria for inclusion of a study in the IPCC meta-analysis. We are still left without a clear idea of what makes this the most up-to-date science.

Reviewer #3 (Remarks to the Author):

The authors addressed several of my comments, but I am not fully convinced by the authors' arguments to support the results of their meta-analysis:

- The authors did not perform a systematic review of the existing literature and, thus, did not follow the usual recommendations made in many meta-analysis guidelines (see, for examples, Nakagawa et al. 2017 BMC Biology 201715:18 or Philibert et al. 2012 Agriculture, Ecosystem and Environment 148:72-82). The submitted paper relies on an unsystematic literature review, and the results are thus not based on the highest existing standards.

- The authors acknowledged in their letter that they did not use all the available published relevant data, especially for wheat. At least, this issue should be addressed in the discussion of the paper and the existence of more comprehensive datasets should be explicitly acknowledged (see my previous review).

- In the rebuttal letter, the justification of the statistical model used by the authors (eq.1) is rather

vague (e.g., "it is derived from theory"). The authors mentioned that some of the included terms may not be selected by model selection methods and explained that they did not include other terms due to the use of a limited dataset, but the authors did not provide clear empirical evidence to support their decisions of including/excluding variables in their statistical model.

- The authors relied on a bootstrap method to analyse model uncertainty. This is indeed useful, but the chosen method deals with one type of uncertainty only (uncertainty in parameter estimates). This is important, but not sufficient; the authors should also analyse the distribution of the residuals (the epsilon in eq.1). In particular, they should check for the existence of bias (e.g., bias resulting from the use of an inappropriate response function), and estimate the residual variance. If a bias is detected (e.g., by checking the distribution of residuals), other response functions may be used. If this variance is large, the residual distribution should be included in the uncertainty analysis.

Referee report for NCOMMS-16-23314A
**“New Science of Climate Change Impacts on Agriculture Implies Higher
Social Cost of Carbon”**

I have been asked to substitute in for Referee 1 (R1). As such, I have carefully read R1’s comments and the authors reply as well as the revised manuscript. In general, I am satisfied with how the authors have dealt with R1’s comments. Below I detail one outstanding issue in the exchange between R1 and the authors that needs further addressing. I also offer several of my own comments that are separate from R1’s critiques.

1 Authors’ reply to R1

R1 Comment (1) remains valid. R1 argues the paper’s central claim that the latest evidence on climate impacts in agriculture necessarily implies a larger social cost of carbon (SCC) in a standard Integrated Assessment Model (i.e. FUND), is not entirely justified.¹ I agree. Is the difference between this paper’s and FUND’s SCC due to outdated studies of the biophysical impacts of temperature on yields? Or is it because of how such biophysical impacts are mapped onto region-specific economic damages (i.e. welfare) via some model of economic reallocation (i.e. trade)? In their response, I think the authors have made a good-faith effort at answering these questions. However, ultimately, definitive answers may be impossible given the murkiness of where the original FUND agricultural damage functions come from.

In particular, the authors note that FUND appears to “reference” earlier versions of GTAP as the model of reallocation such that whatever goes into FUND’s damage function is likely based on a model of reallocation with the same basic structure and underlying data as used in this manuscript. This response, however, is not fully satisfactory. Pages 1-2 of the authors response documents the history of how GTAP was used in earlier versions of FUND. However, the target here is not a past version of FUND, but rather specifically FUND version 3.8.² Was GTAP used in FUND version 3.8? Second, just because the GTAP model was used doesn’t necessarily mean that the same GTAP parameter values were used in FUND 3.8 as was employed by this manuscript. Different GTAP parameter values can lead to different damage parameter estimates. Here, I find the authors’ work showing the spread of damage functions that include GTAP parameter uncertainty to be unconvincing. While they note that “variation of the economic parameters has a modest impact on the underlying uncertainty” (P22L450), Figures S12 and S13 seem to suggest otherwise. The uncertainty associated with GTAP model parameters are quite wide and

¹Summarized in paper’s title and in P13L241 as “Here we show that the current science of climate change impacts on agriculture implies larger damages to the sector than currently represented in models used to calculate the SCC.”

²Or 3.9 according to reference 24 of the manuscript. This should be consistent.

appear to encompass the mean FUND damage function in Figure S12 in nearly every region (though it would be useful to see Figures S12 and S13 overlaid on a common axis to confirm this).

In short, the murkiness of FUND's inner workings makes it difficult to determine the source of why SCC values differ. That means one cannot definitely say that the paper's higher implied SCC is necessarily due to updated biophysical impacts as opposed to differences in economic reallocation.

I don't think this issue necessarily weakens the paper. In some ways the paper's contribution is bigger than just having updated evidence of climate impacts in agriculture, but rather it shows how IAMs can be more transparent in their integration of natural science and economics (i.e. by explicitly showing how biophysical impacts can be inserted into a model of trade). In light of this, here are some suggested changes:

- Change the title to something like: “Updated Science and Modeling of Climate Change Impacts on Agriculture Implies Higher Social Cost of Carbon”
- Be more explicit in your introduction that besides incorporating the latest biophysical damages in agriculture that a major contribution here is your explicit and well-documented use of a model of economic reallocation. This could be its own paragraph rather than the quick mention of “end-to-end” at the end of the introduction.
- Throughout the text, when stating your main result of a higher SCC, please add some caveat along the lines of “it is possible that this higher SCC is due to differences in how economic damages are computed from biophysical impacts.”
- Unless you show additional evidence other than Figures S12 and S13, throughout the text, I would play down that your results are robust to uncertainties related to economic modeling. You note this in footnote 4 on P13 and in P13L246 and P22L450.

2 Additional comments

- (A) P3L65-68: I think I get what you're saying in these three steps but I find it could be more clear. In the first sentence, what do you mean by “aggregated and translated into a consistent set of global impacts”? By aggregation do you mean aggregating across studies or do you mean aggregated spatially or sectorally? Consistent along what? In the second sentence, a non-econ reader will have no idea what you mean by “welfare consequences”. Can you say something like “Then, one needs to map biophysical impacts to economic welfare, which involves modeling both how economies trade and how consumers value consumption of the impacted good.”

- (B) Figure 1: I'm puzzled by the 1995-2005 baseline? Is this a global or country-specific baseline?
- (C) P5L107-109: Related to the above point, I'm not sure how Figure 1 shows "warming consistently more damaging in places that are already hot". Figure 1 shows marginal effects from a change in temperature. I believe the baseline historical temperature is differenced out such that a 1 degree increase in Ecuador is the same as a 1 degree increase in Sweden. How is this claim of varying marginal effects as a function of the historical climate justified?
- (D) P6L125 and elsewhere: I would be more careful regarding how you talk about "statistical uncertainty" in your temperature-yield response functions. We typically think of statistical uncertainty as arising from sampling variability. That's not what's going on here. The uncertainty you're documenting in your temperature-yield function is due to differences in point estimates across studies in Chalinor et al. Those differences are likely due to varying modeling assumptions employed by researchers. Thus the variability you're showing captures discrepancies in how the temperature-yield function is modeled and is akin to the variability across GCM models in climate science. This is fine, but you should be explicit about what you mean by "uncertainty".
- (E) P15-16: It took me a while to realize that Eq. 1 was estimated using historical and not projected data. Please make this explicit in your notation.
- (F) P17L330-332: You define adaptation as representing "changing management practices that would improve yields even in the current climate". This is not the standard definition of adaptation, which is about how practices may change in response to a change climate. Your definition here seems more akin to the "yield-gaps" literature asking why we may see differences in yields across locations with the same climatic features. Please make that explicit.

Reviewer #2 (Remarks to the Author):

First, I want to commend the authors on all of the effort put into the revision of this paper. It does reclaim a lot of the inner workings from a type of “black-box” territory, which was the general theme of my previous comments. In particular, I found the supplementary materials on adaptation to be helpful. I still, however, find that the overall approach may not be good guidance for how to use new evidence to update, though the set of methods for this particular problem and data are more open in this revision.

We thank the reviewer for their positive comment on the revised manuscript. We acknowledge that each economic sector is likely to be very different in terms of the data available and the economic analysis required to produce a damage function, and therefore best practices may differ substantially from sector to sector. We note that generalized guidance is provided in the recent National Academy of Science report on the SCC and that we believe our methodology to be fully consistent with these guidelines (as noted in our second footnote).

One general comment is to make an effort to remove any remaining unsubstantiated negativity about the current fleet of IAMs (as referee 1 pointed to frequently).

We have removed any remaining unsubstantiated negativity. Any comments regarding the current set of IAM damage functions are fully supported, either with reference to the model's own documentation or to other peer reviewed publications.

I present additional comments to the current draft below in no particular order.

1. The comment about 20 year old science should be removed from the abstract. The first papers in the current metaanalysis are also 20 years old. Again, it is not about recency, but the fact that they have been superseded by other results and they are not transparent.

We thank the reviewer for this comment and agree that the principle issue is not the date of studies used for calibration, so much that the details of the calibration is difficult to trace and that dated studies, at least in this sector, do not reflect current understanding of impacts.

We have revised the sentence in the abstract to more specifically reflect this concern: “Despite substantial advances in our understanding of climate change impacts, the scientific basis for damages in the economic models used to calculate the social cost of carbon (SCC) is either undocumented, difficult to trace, or based on a small number of dated studies that may have been superseded by more recent research.”

We have also modified a related sentence in the first paragraph of the introduction: “In most cases the scientific basis for damage functions (reduced-form expressions for how climate change affects economic welfare) in these models is either undocumented, tautological (based on damages from previous versions of the models) or dates from between 10 and 20 years ago and therefore may have been superseded by more recent results.²”

2. If this is being pitched as a general issue / approach, with an underlying scientific support from the IPCC, then some mention should be made of the evidence in the IPCC on other sectors.

We have added the following footnote in the relevant section of the introduction: “Although the meta-analysis from the food security chapter of the most recent IPCC report may have results that can most easily be incorporated into damage functions, other chapters have results that might also be relevant such as those on potential biodiversity loss (Chapter 4), changing fisheries catch (Chapter 6), and changes in key reasons for concern with temperature (Chapter 19).⁵²”

3. In the regression, noting that there are 28 models, the standard errors should probably be clustered on model rather than on study. At least as a robustness check.

We have performed the block bootstrap by model as a robustness check and included it in the supplementary material. We refer to it in the relevant part of the Methods section (p. 16-17): "Figure S16 shows response curves with standard errors based on a model block bootstrap as a robustness check. These are qualitatively similar to the error bars shown in Figure 1, particularly for warming less than 3°C that is the focus of the economic analysis, suggesting the study block bootstrap is capturing the bulk of residual covariance."

We note that the largest effect of this change is a widening of the lower bound of the confidence intervals for some crops, particularly at higher levels of warming (as discussed in the figure caption). This means the effect on the error bars for the SCC would be asymmetric – it would raise the upper bound of the confidence interval, but would not lower the lower bound and therefore would not change our conclusion that the value should be higher than currently represented in FUND.

4. The paper has no citation to Houser et al (2014) (Economic Risks of Climate Change in America), which appears to have taken an empirical approach to damage function estimation not dissimilar to the current paper, though for multiple sectors. This should be included.

References to both Houser et al (2014) and the related more recent paper Hsiang (2017) have been included in relevant places in the introduction.

5. My previous comment 3 asked whether any of the adaptations had been observed in historical data. As a clarification, I was asking this specifically for the adaptations in the process models. Expanding on that, I wish to know how close to reality the adaptation parameters are within those models, not just if a similar adaptation has been observed. In my experience, many process models overstate the benefits from these choice parameters.

To our knowledge there has been no comparison between observed adaptation rates or benefits in agriculture and the parameterization of adaptation in process-based crop models.

6. On the quality of the evidence, more detail should be given on the criteria for inclusion of a study in the IPCC meta-analysis. We are still left without a clear idea of what makes this the most up-to-date science.

On this point we are limited by the description of the methods included in Challinor et al (2016). We include an additional sentence briefly summarizing their Methods and referring readers to the paper (Methods, p.15): "Methods for sampling the literature and criteria for inclusion are described in Challinor et al (2014) as a "broad and inclusive literature search" combined with quality control procedures documented in the Supplemental Information of that paper."

In addition, in response to a related concern from Reviewer 3, we also perform a robustness check where we incorporate 306 additional point estimates for wheat yields from a more recent literature review for that one crop (Wilcox & Makowski, 2014). While this almost doubles the number of observations for wheat, it does not change our estimated response function, as shown in Figure S14.

To our knowledge, the Challinor et al. (2014) analysis and the AgMIP GGCM are the most recent (and some of the only) global, multi-crop, multi-model, estimates of how crop productivity changes with climate change. Though some more recent studies may have been published for individual crops or individual locations, these could only be incorporated through an original literature review for all four crops, a major undertaking well outside the scope of this study.

Reviewer #3 (Remarks to the Author):

The authors addressed several of my comments, but I am not fully convinced by the authors' arguments to support the results of their meta-analysis:

- The authors did not perform a systematic review of the existing literature and, thus, did not follow the usual recommendations made in many meta-analysis guidelines (see, for examples, Nakagawa et al. 2017 BMC Biology 201715:18 or Philibert et al. 2012 Agriculture, Ecosystem and Environment 148:72-82). The submitted paper relies on an unsystematic literature review, and the results are thus not based on the highest existing standards.

An original literature review for all four of the crops included in this study is a major undertaking and well beyond the scope of this study. The goal of this paper is instead to build on existing results on climate impacts on crop productivity and to ask what they imply for economic welfare and the social cost of carbon. We note that the Challinor et al. review has supported results in two peer-reviewed publications (the Nature Climate Change article and the IPCC food security chapter) and therefore believe it is legitimate to use for this purpose.

To address the reviewer's concern, we have:

- 1) Added a sentence describing the literature review methods documented in the Methods section of Challinor et al and referring readers to further documentation in that paper (Methods, p.15):
"Methods for sampling the literature and criteria for inclusion are described in Challinor et al (2014) as a "broad and inclusive literature search" combined with quality control procedures documented in the Supplemental Information of that paper."
- 2) Added a robustness check that includes additional studies from the Wilcox and Makowski review referenced by the reviewer (described more fully in response to the comment below).

- The authors acknowledged in their letter that they did not use all the available published relevant data, especially for wheat. At least, this issue should be addressed in the discussion of the paper and the existence of more comprehensive datasets should be explicitly acknowledged (see my previous review).

In response to this concern, we have performed a robustness check where we merge studies included in Wilcox and Makowski but not in Challinor et al. into the Challinor et al. database. We then re-estimate Equation 1 and compare the estimated wheat yield response curve with and without the additional point estimates in Figure S16. The observations from Wilcox and Makowski that can be incorporated are limited to those that contain values for all variables in Equation 1, and these are merged with baseline temperatures for the growing season and growing areas of the relevant country for the study. This adds 306 point estimates, almost doubling the sample for wheat yields but does not change the estimated response curve, as shown in Figure S16.

The existence of a more recent and more systematic literature review for wheat yields is noted in the Methods section, with reference to this analysis and Figure S16 (Methods, p.15): "Recognizing the existence of a more recent and possibly more systematic literature review for wheat yields, we perform a robustness check where we incorporate additional results identified in Wilcox and Makowski³². This

substantially increases the number of observations for wheat, but does not affect our estimated response curve (Figure S16).”

The figure and caption are given below:

Figure S16: Comparison of the wheat yield response curve for the median baseline temperature estimated using Equation 1 based on the database in Challinor et al¹⁶ and based on an expanded database that adds 6 studies included in Wilcox and Makowski³² but not Challinor et al¹⁶. This is the subset of observations that include data for all variables necessary to estimate Equation 1 (change in temperature, rainfall, CO₂, and whether or not the study included adaptation). The observations from the Wilcox and Makowski database are combined with baseline growing-season temperature for growing areas in the relevant country as described in the Methods section for the Challinor et al. database. These additions increase the number of point estimates of wheat yield from 336 to 642 and the total number of point estimates from 1010 to 1316, but do not substantively change the estimated response function. The confidence interval is the 95% interval based on the block bootstrap of the regression using just the Challinor et al. database.

- In the rebuttal letter, the justification of the statistical model used by the authors (eq.1) is rather vague (e.g., “it is derived from theory”). The authors mentioned that some of the included terms may not be selected by model selection methods and explained that they did not include other terms due to the use of a limited dataset, but the authors did not provide clear empirical evidence to support their decisions of including/excluding variables in their statistical model.

We acknowledge that our comment in the previous revision that the empirical model was derived from theory was incomplete and vague. More specifically, this comment reflects the fact that at a minimum, temperature damage functions in agriculture should:

- 1) Allow the effect of warming to differ by crop
- 2) Allow the effect of warming to be non-linear
- 3) Allow the effect of warming to be different in hot vs cold places

- 4) Account for the effect of CO₂ fertilization, including that this might differ between C3 and C4 crops
- 5) Control for changes in rainfall
- 6) Account for the effects of on farm adaptation

The table below gives the results of Wald tests that compare our model to restricted models that sequentially remove each of the 6 characteristics described above (standard errors were clustered at the study level for this analysis, rather than using the non-parametric block bootstraps, because these model comparison tests require parameterization of model variance):

Terms Removed from Regression	F-Statistic	Probability Restricted Model = Unrestricted Model
1. All crop interaction terms	11.24	<2e-16
2. All quadratic warming terms	6.083	1.2e-7
3. All interaction terms between warming and baseline temperature	14.67	<2e-16
4.1 All CO ₂ fertilization terms	22.47	2.9e-10
4.2 CO ₂ fertilization just for C4	37.57	1.27e-9
5. Rainfall Control	10.86	0.001
6. All adaptation terms	2.57	0.07

There is strong evidence that all terms add explanatory power to the model, with the slight exception of the adaptation terms. These terms nevertheless have to be included because economic theory requires that climate damage functions account for the benefits of adaptation. Our model is therefore the most parsimonious that meets these requirements. Parsimony is desirable because of the limited number of studies included and the fact that this set of parameters alone results in 21 estimated parameters from 56 studies in the database.

We also perform Wald tests of additional model terms. We test whether a squared precipitation term should be added and the null hypothesis that this additional term is zero is not rejected ($p=0.57$).

We also test the inclusion of cubic warming terms, allowing the parameters to differ by crop and by baseline temperature in the same way as the linear and quadratic terms. Though there is evidence that these terms add explanatory power to the model ($p<0.01$), the temperature response functions do not differ significantly from those estimated without cubic terms, as shown in the figure below (solid line = no cubic terms, dashed line = with cubic terms, 95% confidence interval for the cubic model estimated using a block bootstrap, blocking at the study level).

- The authors relied on a bootstrap method to analyze model uncertainty. This is indeed useful, but the chosen method deals with one type of uncertainty only (uncertainty in parameter estimates). This is important, but not sufficient; the authors should also analyse the distribution of the residuals (the epsilon in eq.1). In particular, they should check for the existence of bias (e.g., bias resulting from the use of an inappropriate response function), and estimate the residual variance. If a bias is detected (e.g., by checking the distribution of residuals), other response functions may be used. If this variance is large, the residual distribution should be included in the uncertainty analysis.

The distribution of residuals for the regression is shown below:

The normal quantile-quantile plot of the standardized residuals is shown below:

Although the distribution shows some deviation from normality in the tails, the residuals are symmetric about zero and the bulk of the distribution is closely approximated by the normal. Given we are using non-parametric methods to estimate our standard errors, we do not believe the deviation from normality shown in the tails of the distribution affects our results.

We also note that the parameter uncertainty is directly connected to the variance of the residuals. For instance, the standard parametric distribution of parameter variance is given by:

$$\text{Var}(\boldsymbol{\beta}) = \sigma^2(\mathbf{X}^T \mathbf{X})^{-1}$$

Where σ^2 is the variance of the regression residuals. Although we are not using this parametric form for our standard errors, a similar relationship between the variance of the parameters and the variance of the residuals exists for the block bootstrap: a regression with small residual variance will give more tightly-estimated parameters and one with large residual variance will produce parameters with higher variance.

Reviewer #4 (Remarks to the Author):

I have been asked to substitute in for Referee 1 (R1). As such, I have carefully read R1's comments and the authors reply as well as the revised manuscript. In general, I am satisfied with how the authors have dealt with R1's comments. Below I detail one outstanding issue in the exchange between R1 and the authors that needs further addressing. I also offer several of my own comments that are separate from R1's critiques.

- We thank the reviewer for agreeing to review the comments from Reviewer 1 and our response, both of which are substantial, and for their thoughtful and constructive comments.

1 Authors' reply to R1

R1 Comment (1) remains valid. R1 argues the paper's central claim that the latest evidence on climate impacts in agriculture necessarily implies a larger social cost of carbon (SCC) in a standard Integrated Assessment Model (i.e. FUND), is not entirely justified.¹ I agree.

Is the difference between this paper's and FUND's SCC due to outdated studies of the biophysical impacts of temperature on yields? Or is it because of how such biophysical impacts are mapped onto region-specific economic damages (i.e. welfare) via some model of economic reallocation (i.e. trade)? In their response, I think the authors have made a good-faith effort at answering these questions. However, ultimately, definitive answers may be impossible given the murkiness of where the original FUND agricultural damage functions come from.

In particular, the authors note that FUND appears to "reference" earlier versions of GTAP as the model of reallocation such that whatever goes into FUND's damage function is likely based on a model of reallocation with the same basic structure and underlying data as used in this manuscript. This response, however, is not fully satisfactory. Pages 1-2 of the authors response documents the history of how GTAP was used in earlier versions of FUND. However, the target here is not a past version of FUND, but rather specifically FUND version 3.8.2 Was GTAP used in FUND version 3.8?

- The calibration of the agricultural damage function in FUND has not been updated since the model was initially developed. All our discussion in the previous response to reviewers on the relationship between previous CGE models used in the existing calibration and in the current paper was based on documentation from the current version of FUND.

Second, just because the GTAP model was used doesn't necessarily mean that the same GTAP parameter values were used in FUND 3.8 as was employed by this manuscript. Different GTAP parameter values can lead to different damage parameter estimates. Here, I find the authors' work showing the spread of damage functions that include GTAP parameter uncertainty to be unconvincing. While they note that "variation of the economic parameters has a modest impact on the underlying uncertainty" (P22L450), Figures S12 and S13 seem to suggest otherwise. The uncertainty associated with GTAP model parameters are quite wide and appear to encompass the mean FUND damage function in Figure S12 in nearly every region (though it would be useful to see Figures S12 and S13 overlaid on a common axis to confirm this).

- We believe the reviewer may have mis-read the two sets of error bars shown in Figure S13 (original figure copied below, along with the figure caption). The large error bars on the left gives variance associated with the parameters in the meta-analysis (i.e. uncertainty associated with how crop yields respond to temperature). The much smaller error bars to the right of these large ones show the variance associated with the GTAP sensitivity analysis.
- We believe the reviewers comments refer to the former, while our comments regarding the insensitivity of results to GTAP parameters refer to the latter. The former set of error bars are essentially the same as those shown in Figure 3 (though not precisely the same because of the lower resolution that GTAP was run at for the sensitivity analysis). Figure 3 shows these error bars together with the existing FUND damage functions on the same scale.

Figure S13: Results of the GTAP sensitivity analysis (Methods, Table S7) run using a 16-region version of GTAP (matching the FUND regions). The two sets of error bars show the 95% confidence interval based on uncertainty in the yield response to temperature (larger error bars, left hand side) and +/- 2 standard deviations based on a sensitivity analysis of key parameters in GTAP (smaller error bars, right hand side). Because these damage functions come from a more aggregated version of GTAP, they are slightly different from those shown in Figure 3 and Figure S12 that come from the full 140-region version of the model.

- The idea behind including both sets of error bars in Figure S13 was to directly show the relative importance of the two types of uncertainty. In retrospect though it is clearly confusing, particularly given the very different magnitudes of the two sets of error bars. Therefore, in the revised manuscript we have replaced Figure S13 with a version showing error bars only from the GTAP sensitivity analysis.
- The small error bars in the revised Figure S13 clearly show the very small sensitivity of results to parametric uncertainty in the economic modeling step, and therefore how unlikely it is that the differences we document between damage functions is a result of changing economic models.

In short, the murkiness of FUND's inner workings makes it difficult to determine the source of why SCC values differ. That means one cannot definitely say that the paper's higher implied SCC is necessarily due to updated biophysical impacts as opposed to differences in economic reallocation.

- We agree that absent introducing our revised biophysical impacts into the CGE and PE models initially used to calibrate FUND, one can not *conclusively* rule out changing economic models have affected the difference in damages that we document.
- However, we believe it is highly unlikely to be a determining factor for several reasons:

- The structure of the economic model we used is related to that in models used in for the existing FUND calibration, as documented in the previous response to reviewers
- Our sensitivity analysis shows very little variation in damages in response to plausible variation on a range of important GTAP parameters
- Within our results, damages are clearly sensitive to different biophysical shocks, as shown by the range of our error bars in Figure 3, as well as the difference between the Meta-Analysis, AgMIP (All), and AgMIP (Preferred) results.
- The economic modeling framework permits us to decompose the global welfare effects into three components: the direct impact of lower productivity, the terms of trade (ToT) effect, and the indirect interactions with existing policy distortions (allocative efficiency component). Using this decomposition we can see why economic parameters would not be expected to substantially affect the global SCC:
 1. The direct impacts of productivity changes due to climate are locally approximated by the inner product of the values of regional production and the percentage changes in productivity. Therefore, to a first-order approximation, this depends on the data (derived from the GTAP data base) and the productivity shocks (derived from the meta-analysis). Therefore, it is unsurprising that this component of the welfare change is insensitive to the economic parameters in the model.
 2. The allocative efficiency effects, on the other hand, depend on the interaction between quantity changes in the model and pre-existing distortions. The quantity changes are heavily dependent on economic parameters. However, overall, the size of the allocative efficiency effect is very small compared to the direct productivity effect.
 3. The ToT effect depends on the interactions between commodity prices and a given region's exports and imports. Since the price changes depend on the economic parameters, the ToT effects are also sensitive to variation in these parameters. However, since the ToT effects are purely re-distributive and sum to zero globally, they are unlikely to have a major effect on the global SCC.

I don't think this issue necessarily weakens the paper. In some ways the paper's contribution is bigger than just having updated evidence of climate impacts in agriculture, but rather it shows how IAMs can be more transparent in their integration of natural science and economics (i.e. by explicitly showing how biophysical impacts can be inserted into a model of trade). In light of this, here are some suggested changes:

- Change the title to something like: "Updated Science and Modeling of Climate Change Impacts on Agriculture Implies Higher Social Cost of Carbon"

Given the weight of evidence documented in the four bullets above, we believe this title, by implying that the changing biophysical impacts and the difference in economic model are equally likely to be responsible for the difference in damage we document, would be misleading.

- Be more explicit in your introduction that besides incorporating the latest biophysical damages in agriculture that a major contribution here is your explicit and well documented use of a model of economic reallocation. This could be its own paragraph rather than the quick mention of "end-to-end" at the end of the introduction.

We have expanded our discussion of the three linked parts of the analysis, including the updated economic modeling section at the end of the introduction (p. 5): “Finally, this is an “end-to-end” in which 1) the analysis of underlying biophysical impacts literature to produce global, sector-wide response functions, 2) the modeling of the economic responses to those biophysical changes using a state of the art, open-source CGE model to produce economic damages, and 3) the introduction of these damages into an IAM and the resulting effect on the SCC are all documented within a single study. This means our damage functions and the changes we identify in the SCC have a clear and traceable connection to the underlying science that is both comprehensive and up-to-date, in contrast to most current IAM damage functions.¹”

- Throughout the text, when stating your main result of a higher SCC, please add some caveat along the lines of “it is possible that this higher SCC is due to differences in how economic damages are computed from biophysical impacts.”

We have added the following caveats in the text (additions shown in bold):

Results (p.11): “We use an IAM damage module based on the FUND model and substitute our new agricultural damage functions into the agricultural sector to calculate how the SCC changes as a result of this more up-to-date science, **and new economic modeling** (Methods).”

Discussion (p. 13): “Here we show that the current science of climate change impacts on agriculture, **combined with up-to-date economic modeling**, implies larger damages to the sector than currently represented in models used to calculate the SCC.”

- Unless you show additional evidence other than Figures S12 and S13, throughout the text, I would play down that your results are robust to uncertainties related to economic modeling. You note this in footnote 4 on P13 and in P13L246 and P22L450.

- We hope that given our explanation of the sensitivity analysis results above and our revised Figure S13 more directly showing the small response of damage estimates to parametric uncertainty in GTAP, the Reviewer feels our characterization of these results is reasonable.
- We have moved the discussion of the sensitivity from the original footnote into the main text, for a more thorough discussion of this question: Results (p. 11): “In order to investigate the importance of parameterization of the economic model used to calculate damages in driving the regional damage functions, we perform a systematic sensitivity analysis of key parameters within GTAP (Methods, Table S7). Variation in economic welfare changes associated with these changes is shown in Figure S13 and is very small and dwarfed by the importance of uncertainty in biophysical productivity shocks shown in the error bars in Figure 3.”

2 Additional comments

¹ We also note that a recent review of methods used by the US government to calculate the SCC by the National Academy of Sciences recommended criteria for IAM damage functions including that: 1) they should be consistent with the scientific basis as represented in the current peer-reviewed literature; 2) uncertainties should be characterized and quantified where possible; and 3) the process should be transparent, well-documented, and reproducible. We believe the approach used in this paper to be fully consistent with these criteria².

(A) P3L65-68: I think I get what you're saying in these three steps but I find it could be more clear. In the first sentence, what do you mean by "aggregated and translated into a consistent set of global impacts"? By aggregation do you mean aggregating across studies or do you mean aggregated spatially or sectorally? Consistent along what? In the second sentence, a non-econ reader will have no idea what you mean by "welfare consequences". Can you say something like "Then, one needs to map biophysical impacts to economic welfare, which involves modeling both how economies trade and how consumers value consumption of the impacted good."

- We have revised this section to read:

"For each sector, individual scientific studies must be aggregated and translated into a consistent set of global impacts. Then, because damage functions parameterize how economic welfare changes with temperature, the economic value of these biophysical impacts must be assessed, which might involve substantial economic modeling."

- For the discussion of welfare consequences, this statement is about economic sectors in general and we have therefore kept the statement more general than suggested. For agriculture, modeling trade is important, but for other sectors, calculating the welfare consequences might not necessitate the kind of CGE modeling we undertake in this paper (e.g. mortality).

(B) Figure 1: I'm puzzled by the 1995-2005 baseline? Is this a global or country-specific baseline?

- We thank the reviewer for pointing this out. The relevant baseline in Figure 1 is the local (country-specific) baseline (since this shows the yield response to local temperature change). The baseline in Figures 2 and 3 is the global average though, because the relationship between global and local temperature changes has been parameterized into the regional damage functions. This distinction has been clarified in the revised manuscript.

(C) P5L107-109: Related to the above point, I'm not sure how Figure 1 shows "warming consistently more damaging in places that are already hot". Figure 1 shows marginal effects from a change in temperature. I believe the baseline historical temperature is differenced out such that a 1 degree increase in Ecuador is the same as a 1 degree increase in Sweden. How is this claim of varying marginal effects as a function of the historical climate justified?

- The β_{3j} and β_{4j} terms in the estimating equation (Equation 1) are the interaction between the marginal effect of warming and the baseline average growing season temperature. This allows the effect of 1 degree of warming to differ depending on whether a place was initially hot or initially cold.

- The three lines in each of the four graphs in Figure 1 give the effects of warming for the 25th, 50th, and 75th percentile of growing season temperature. The darkest lines show the effect in the warmest place, and these are consistently more negative than the others.

(D) P6L125 and elsewhere: I would be more careful regarding how you talk about "statistical uncertainty" in your temperature-yield response functions. We typically think of statistical uncertainty as arising from sampling variability. That's not what's going on here. The uncertainty you're documenting in your temperature-yield function is due to differences in point estimates across studies in Chalinor et al. Those differences are likely due to varying modeling assumptions employed by researchers. Thus the variability you're showing captures discrepancies in how the temperature-yield

function is modeled and is akin to the variability across GCM models in climate science. This is fine, but you should be explicit about what you mean by “uncertainty”.’

- We thank the reviewer for pointing this out and agree that the uncertainty we are dealing with here is not the classic sampling variability usually thought of as statistical uncertainty. We have removed the phrase “statistical uncertainty” from the manuscript. In most places when referring to the variation associated with our meta-analysis confidence intervals we have replaced the word “uncertainty” with the word “spread” or “range”, to better capture the fact that this variation is more akin to the spread in a model ensemble than sampling variability.

(E) P15-16: It took me a while to realize that Eq. 1 was estimated using historical and not projected data. Please make this explicit in your notation.

- Equation 1 is estimated using point estimates from studies that perturb temperature, rainfall and CO₂ variables and estimate yield consequences using either a process-based or empirical crop model. In most cases, the way in which the climate variables are perturbed is consistent with projected climate change, but are not necessarily climate projections. Most studies could instead be thought of as sensitivity studies (e.g. how much does yield change for a temperature change of X degrees?). Some studies use projected climate variables associated with climate model output. Equation 1 is not estimated using historical data.

(F) P17L330-332: You define adaptation as representing “changing management practices that would improve yields even in the current climate”. This is not the standard definition of adaptation, which is about how practices may change in response to a change climate. Your definition here seems more akin to the “yield-gaps” literature asking why we may see differences in yields across locations with the same climatic features. Please make that explicit.

- This characterization is not how we define adaptation, but rather a description of most of the actions labeled “adaptation” in the studies in the meta-analysis. This is why we have to include an adaptation intercept term, in addition to what we term the “true” adaptation term, represented by the *interaction* between warming and the inclusion of adaptation in the study.

- In both the methods and results sections we take pains to describe this distinction between the de facto approach to modeling “adaptation” that we observe in our set of studies, and the standard definition of adaptation as actions that improve yields under the future climate but not under the current one.

Results (p. 7): “The effect of agronomic, on-farm, within-crop adaptations (principally changes in crop variety and planting date (Methods)) is small and statistically insignificant. Studies that include agronomic adaptation do, on average, report higher yields than those that don’t, but this is almost entirely captured by an adaptation intercept term rather than the interaction with change in temperature. This suggests changes described as adaptations to climate change in the studies underpinning our meta-analysis would provide similar yield benefits with and without climate change, and are therefore not true climate adaptations as conventionally defined²². In results that follow we include only the true climate adaptation effect for all crops, but we find this to be small (Table S1).”

Methods (p. 18): “The true effect of adaptation is the interaction with temperature change, given by the β_8 term in Equation 1, which is included in all subsequent analyses. This term captures the effect of management changes that are not beneficial today but that will become beneficial under a changed climate, the standard definition of adaptation.”

REVIEWERS' COMMENTS:

Reviewer #2 (Remarks to the Author):

Thank you again for the opportunity to review this paper. I'd like to once more commend the authors on dealing with comments thoroughly and thoughtfully with reviewer responses.

Most of my substantive concerns have been addressed by the current draft. One concern is that the definition and treatment of adaptation remains rather opaque. A short additional note or paragraph discussing current scientific knowledge on adaptation, behaviours observed in practice, and their relationship to the adaptations employed by the agricultural models would be helpful. This would emphasise that the adaptation piece is a substantial weakness of all SCC and climate impact analyses, and provide context for the current treatment.

Reviewer #3 (Remarks to the Author):

The presentation of the meta-analysis was significantly improved. I found the sensitivity analyses performed by the authors interesting; it increases the scientific value of the article. The explanations given by the authors to justify their statistical model are now more convincing.

Still, some aspects could be improved:

- In Table S1, the authors give some estimated values, but not for all parameters. For transparency, the authors should provide all estimated parameter values and, also, the results of their statistical tests (the test results are given in the response letter, but apparently not in the paper itself).
- The choices made by authors in their uncertainty analysis are not fully justified. In their response letter, the authors explained that the variance of the parameter estimator is related to the residual variance. This is true, but it does not mean that the full uncertainty in the predicted yield change is reflected by the variance of the parameter estimator. Indeed, for a statistical regression model relating Y (here, yield change) to X (here, climate inputs), two types of confidence intervals can be derived to analyse uncertainty (see textbooks on regression); (i) the confidence interval for the estimated mean response (reflecting the uncertainty of the estimate of conditional mean $E(Y|X)$), (ii) the confidence interval for specific predicted values (which takes into account the fact that the chosen inputs X do not explain all the variability of Y). The first one is derived from the distribution of the parameter estimates. The second one is derived from the same distribution PLUS the distribution of the residual errors. Both types of confidence interval can be computed using either parametric or non-parametric methods. The second one is larger than the first one, unless X explain all the variability of Y . I understood from the authors' explanations that the first type of confidence interval was used. If so, the authors should justify their choice.

Reviewer #4 (Remarks to the Author):

I am satisfied by this latest round of revisions and have no further comments.

Response to Reviewers

Reviewer #2 (Remarks to the Author):

Thank you again for the opportunity to review this paper. I'd like to once more commend the authors on dealing with comments thoroughly and thoughtfully with reviewer responses.

Most of my substantive concerns have been addressed by the current draft. One concern is that the definition and treatment of adaptation remains rather opaque. A short additional note or paragraph discussing current scientific knowledge on adaptation, behaviours observed in practice, and their relationship to the adaptations employed by the agricultural models would be helpful. This would emphasise that the adaptation piece is a substantial weakness of all SCC and climate impact analyses, and provide context for the current treatment.

- We have added the following statement to the relevant part of the supplementary information (caption of Supplementary Table 2):

“Note that although adaptive behaviors by farmers have been documented in real-world settings⁵⁻⁷, consistent with some of the adaptations represented in process-based crop models, no empirical validation of the magnitude of these benefits has been undertaken.”

- We also note that adaptation as represented in process-based models is found to have a very minor effect on our results. We therefore don't believe that, in this specific case, it is a substantial weakness of the SCC calculation and believe that an extended discussion of the issue in the main text would be out of place.

Reviewer #3 (Remarks to the Author):

The presentation of the meta-analysis was significantly improved. I found the sensitivity analyses performed by the authors interesting; it increases the scientific value of the article. The explanations given by the authors to justify their statistical model are now more convincing.

Still, some aspects could be improved:

- In Table S1, the authors give some estimated values, but not for all parameters. For transparency, the authors should provide all estimated parameter values and, also, the results of their statistical tests (the test results are given in the response letter, but apparently not in the paper itself).
- Coefficients for complex, non-linear functions are not directly interpretable and are therefore the temperature terms estimated from Equation 1 are not reported in Table S1. Moreover, significance tests for individual coefficients in a polynomial function are misleading because the over all significance of the variable depends on the covariance between the parameters (e.g. between the coefficients on T and T^2), not just on the variance of the individual parameters. It is clearer and more accurate to show combined effect of all parameters, as well as their joint significance. This is given in Figure 1.

- Results of the robustness checks included in the previous response to reviewers have been added to the Supplementary Information (Supplementary Table 7 and Supplementary Figure 20).
 - The choices made by authors in their uncertainty analysis are not fully justified. In their response letter, the authors explained that the variance of the parameter estimator is related to the residual variance. This is true, but it does not mean that the full uncertainty in the predicted yield change is reflected by the variance of the parameter estimator. Indeed, for a statistical regression model relating Y (here, yield change) to X (here, climate inputs), two types of confidence intervals can be derived to analyse uncertainty (see textbooks on regression); (i) the confidence interval for the estimated mean response (reflecting the uncertainty of the estimate of conditional mean $E(Y|X)$), (ii) the confidence interval for specific predicted values (which takes into account the fact that the chosen inputs X do not explain all the variability of Y). The first one is derived from the distribution of the parameter estimates. The second one is derived from the same distribution PLUS the distribution of the residual errors. Both types of confidence interval can be computed using either parametric or non-parametric methods. The second one is larger than the first one, unless X explain all the variability of Y . I understood from the authors' explanations that the first type of confidence interval was used. If so, the authors should justify their choice.
- The following sentence has been added in the methods section: "All error bars reported in the paper show confidence intervals rather than prediction intervals. This is appropriate since the relevant uncertainty is in the expected response of yield to temperature change, which is given by confidence intervals."

Reviewer #4 (Remarks to the Author):

I am satisfied by this latest round of revisions and have no further comments.